# Cold cloud microphysical process rates in a global chemistry-climate model

Sara Bacer[1,*], Sylvia C. Sullivan[2], Odran Sourdeval[3], Holger Tost[4], Jos Lelieveld[1,5], and Andrea Pozzer[1]

[1]Atmospheric Chemistry Department, Max Planck Institute for Chemistry, Mainz, Germany
[2]Institute of Meteorology and Climate Research, Karlsruhe Institute of Technology, Karlsruhe, Germany
[3]Laboratoire d'Optique Atmosphérique, Université de Lille, CNRS, Lille, France
[4]Institute for Atmospheric Physics, Johannes Gutenberg University Mainz, Mainz, Germany
[5]Climate and Atmosphere Research Center, The Cyprus Institute, Nicosia, Cyprus
[*]*now at:* LEGI, Université Grenoble Alpes, CNRS, Grenoble INP, Grenoble, France

**Correspondence:** Sara Bacer (sara.bacer@univ-grenoble-alpes.fr)

**Abstract.** Microphysical processes in cold clouds which act as sources or sinks of hydrometeors below $0°C$ control the ice crystal number concentrations (ICNCs) and in turn the cloud radiative effects. Estimating the relative importance of the cold cloud microphysical process rates is of fundamental importance to underpin the development of cloud parameterizations for weather, atmospheric chemistry and climate models and compare the output with observations at different temporal resolutions.

This study quantifies and investigates the ICNC rates of cold cloud microphysical processes by means of the chemistry-climate model EMAC and defines the hierarchy of sources and sinks of ice crystals. Both microphysical process rates, such as ice nucleation, aggregation, and secondary ice production, and unphysical correction terms are presented. Model ICNCs are also compared against a satellite climatology. We found that model ICNCs are in overall agreement with satellite observations in terms of spatial distribution, although the values are overestimated, especially around high mountains. The analysis of ice

crystal rates is carried out both at global and at regional scales. We found that globally the freezing of cloud droplets and convective detrainment over tropical land masses are the dominant sources of ice crystals, while aggregation and accretion act as the largest sinks. In general, all processes are characterised by highly skewed distributions. Moreover, the influence of (a) different ice nucleation parameterizations and (b) a future global warming scenario on the rates has been analysed in two sensitivity studies. In the first, we found that the application of different parameterizations for ice nucleation changes the

hierarchy of ice crystal sources only slightly. In the second, all microphysical processes follow an upward shift in altitude and an increase by up to 10% in the upper troposphere towards the end of the 21st century.

## 1 Introduction

Clouds play a central role in the global energy budget interacting with shortwave solar and longwave terrestrial radiation. Their radiative properties (cloud albedo and emissivity) depend on microphysical and optical characteristics, such as temperature,

size distribution and shape of cloud particles, and the phase of water. Despite their important role in the Earth System, the understanding of clouds is still challenging and affected by large uncertainties (IPCC, 2013). The numerical representation

of clouds must contend with the limited understanding of the fundamental details of microphysical processes as well as the fact that cloud processes span several order of magnitudes (from nanometres to thousands of kilometres). Hence, modelling of clouds remains a weak point in all atmospheric models, regardless of their resolution, and has been recognised as one of the dominant sources of uncertainty in climate studies (IPCC, 2013; Seinfeld et al., 2016).

Modelling the microphysics of cold clouds, which form at temperatures lower than $0°C$ and involve ice crystals (ICs), is more challenging than that of warm clouds because of the additional complexity of ice processes (Cantrell and Heymsfield, 2005; Kanji et al., 2017; Heymsfield et al., 2017; Korolev et al., 2017; Dietlicher et al., 2019). Some examples of these processes include heterogeneous ice nucleation, which depends on particular aerosols and occurs via different modes; the secondary production mechanisms of ice crystals, which involve collisions of ICs; the competition for water vapour among different ice particles; and the thermodynamic instabilities when liquid and ice phases coexist. Additionally, the variety of possible ice crystal shapes from dendrites to needles also determines the radiative impact of cold clouds and complicates their representation in large-scale models (Lawson et al., 2019). Cold clouds are classified as cirrus clouds when they purely consist of ICs at temperatures generally lower than $-35°C$, and as mixed-phase clouds when they include both ICs and supercooled liquid cloud droplets between $-35°C$ and $0°C$. Cirrus clouds strongly impact the transport of water vapour entering the stratosphere, which in turn has a strong effect on radiation and ozone chemistry (Jensen et al., 2013), and produce a positive net radiative effect at the top of the atmosphere (TOA) (Chen et al., 2000; Hong et al., 2016; Matus and L'Ecuyer, 2017); on the other hand, mixed-phase clouds exert a negative net radiative effect at the TOA, although the estimates of their radiative effect are complicated by the coexistence of both ice and liquid cloud phases (Chen et al., 2000; Hong et al., 2016; Matus and L'Ecuyer, 2017).

Several categories of microphysical processes have been identified in cold clouds (Pruppacher and Klett, 1997). These can be broadly classified as formation, growth, and loss processes of ice crystals. New ICs are formed thermodynamically via two ice nucleation mechanisms, depending on environmental conditions (e.g. temperature, supersaturation, and vertical air motions) and aerosol populations (i.e. aerosol number concentrations and physicochemical characteristics, such as composition, shape, and surface tension) (Pruppacher and Klett, 1997; Kanji et al., 2017; Heymsfield et al., 2017). Homogeneous ice nucleation occurs at low temperatures (below $-35°C$) and high ice saturation ratios ($140\% - 160\%$) via the freezing of supercooled liquid cloud droplets. Heterogeneous ice nucleation takes place at warmer but subzero temperatures and lower ice supersaturation thanks to the presence of particular atmospheric aerosols, called ice nucleating particles (INPs). It occurs via four different mechanisms, or ice nucleation modes: contact nucleation, condensation nucleation, immersion, and deposition nucleation modes. ICs can also be produced from the multiplication of pre-existing ice crystals, via the so-called secondary ice production (or ice multiplication). Several mechanisms of secondary ice production have been identified. In rime splintering (or the Hallett-Mossop process), small ice crystals (or splinters) are ejected after the capture of supercooled droplets by large ice particles (e.g. graupels) between $-3°C$ and $-8°C$. In collisional break-up (or collisional fragmentation), the disintegration of fragile, slower-falling dendritic crystals that collide with dense graupel particles produces smaller ice particles. Droplet shattering involves the freezing of large cloud droplets and their subsequent shattering. Sublimation fragmentation occurs when ice

particles break from parent ice particles after the sublimation of "ice bridges" at ice subsaturated conditions. Additionally, ICs can be generated in the vicinity of deep convective clouds by their lateral outflow or detrainment.

A variety of ice growth mechanisms also exist. In conditions of ice supersaturation, ICs grow by diffusion as ambient water vapour deposits. When both ice and liquid phases coexist, the water vapour is generated by evaporating water droplets because
of the difference between the saturation vapour pressure over ice and over water (Wegener-Bergeron-Findeisen – WBF mechanism). The collision-coalescence (or collection) between ICs and other hydrometeors is another growth mechanism which occurs in several ways (Rogers and Yau, 1989; Khain and Pinsky, 2018): self-collection consists of the collision-coalescence between ICs and the production of ice crystals with larger size; aggregation occurs when the colliding ICs clump together to form an aggregated snowflake; accretion indicates the collection between ice crystals and snowflakes; and riming refers to the
collision of ICs with supercooled liquid droplets which freeze upon contact. Melting and sublimation are other sinks of ice crystals when temperatures are higher than 0°C and there is ice subsaturation, respectively.

Ice crystal number concentration (ICNC) influences microphysical and optical properties of cold clouds, so an accurate ICNC estimate allows for a more realistic representation of the cloud radiative effects. Many efforts have been made to parameterize all relevant microphysical processes which affect ICNC (e.g. Kärcher and Lohmann, 2002a; Barahona and Nenes, 2008;
Phillips et al., 2007; DeMott et al., 2010; Hallett and Mossop, 1974) and to further improve the existing parameterizations (e.g. Kärcher and Lohmann, 2002b; Barahona and Nenes, 2009; Phillips et al., 2013; DeMott et al., 2016; Sullivan et al., 2018b, a). The parameterizations have been implemented in general circulation models (GCMs) which may use a two-moment cloud microphysics scheme (e.g. Liu et al., 2012; Barahona et al., 2014; Kuebbeler et al., 2014; Bacer et al., 2018) to advance the simulation of cloud phase partitioning and cloud-radiation feedbacks.

It is of crucial importance to know the hierarchy of sources and sinks of ICs under different thermodynamic conditions and over different time scales. In fact, knowing these relative contributions facilitates the comparison of simulation output with observations across temporal resolutions and the development of scale-aware microphysics schemes. Gettelman et al. (2013) analysed the rates of the processes affecting precipitation in the CAM5 model. Muench and Lohmann (2020) presented some information about ice crystal sources in the ECHAM-HAM model. Nevertheless, to the best of our knowledge, a detailed
quantitative analysis of all the microphysical processes affecting ICNC has not yet been performed. Moreover, ICNC in GCMs is also affected by unphysical correction terms (or numerical rates) that are usually neglected in scientific investigations. Therefore, this study aims to estimate and investigate carefully the rates of the microphysical processes and the unphysical corrections which act as sources or sinks of ice crystals and control ICNC in cold clouds for the first time. The analysis is carried out both at global and at regional scales. We also discuss how the rates will change under a global warming scenario
towards the end of the century. For this study, the numerical simulations have been performed with the global ECHAM/MESSy Atmospheric Chemistry (EMAC) model.

The paper is organised as follows. We first describe the EMAC model and the numerical representation of the ICNC rates inside the model (Section 2). Then, the simulations are detailed (Section 3) and the ICNC output data are compared with ICNC satellite estimations (Section 4). The model results for microphysical and numerical rates are presented at both the global and

the regional scale (Section 5); we also show the robustness of these results to the ice nucleation parameterization, as well as their sensitivity to global warming with an RCP6.0 simulation. Finally, we present our conclusions (Section 6).

## 2 Ice microphysical processes in EMAC

### 2.1 The EMAC model

The EMAC model is a global chemistry-climate model which describes tropospheric and middle-atmospheric processes and their interactions with ocean, land, and human influences. EMAC combines the 5th generation European Centre Hamburg GCM (ECHAM5, Roeckner et al., 2006), the core of the atmospheric dynamics computations, with the Modular Earth Submodel System (MESSy, Jöckel et al., 2010), which includes a variety of submodels describing physical, dynamical, and chemical processes. For the present study we used ECHAM5 version 5.3.02 and MESSy version 2.53.

The EMAC model has been extensively used and evaluated against in-situ, aircraft, and satellite observations of, for example, aerosol optical depth, acid deposition, meteorological parameters, cloud radiative effects (e.g. Pozzer et al., 2012, 2015; Karydis et al., 2016; Tsimpidi et al., 2016; Klingmüller et al., 2018; Bacer et al., 2018). EMAC computes gas-phase species online through the Module Efficiently Calculating the Chemistry of the Atmosphere (MECCA) submodel (Sander et al., 2011) and provides a comprehensive treatment of chemical processes and dynamical feedbacks through radiation (Dietmüller et al., 2016). Aerosol microphysics and gas/aerosol partitioning are calculated by the Global Modal-aerosol eXtension (GMXe) submodel (Pringle et al., 2010), a two-moment aerosol module which predicts the number concentration and the mass mixing ratio of the aerosol modes. The aerosol size distribution is described by seven lognormal modes: four hydrophilic modes, which cover the aerosol size spectrum of nucleation, Aitken, accumulation, and coarse particles, and three hydrophobic modes, which have the same size range except for the nucleation particles. The aerosol composition within each mode is uniform (internally mixed) but varies among the modes (externally mixed). The ONEMIS and OFFEMIS submodels describe the online and offline emissions, respectively, of tracers and aerosols, while the TNUDGE submodel performs the tracer nudging towards observations (Kerkweg et al., 2006b). Physical loss processes, like dry deposition, wet deposition, and sedimentation of aerosols and trace gases, are explicitly considered by the submodels DDEP, SEDI, and SCAV (Kerkweg et al., 2006a; Tost et al., 2006a). The RAD submodel (Dietmüller et al., 2016) calculates the radiative transfer taking into account cloud cover, optical properties of clouds and aerosols, mixing ratios of water vapour and radiatively active species, and orbital parameters. Convective and large-scale clouds are parameterized via two different submodels, the CONVECT submodel (Tost et al., 2006b) and the CLOUD submodel (Roeckner et al., 2004), as described in the next Subsection.

In EMAC, a single updraft velocity ($w$) is used for the whole grid cell, although the vertical velocity varies strongly in reality within the dimensions of a grid box (e.g. Guo et al., 2008). This is a simplification which is commonly used by GCMs. The subgrid-scale variability of vertical velocity ($w_{sub}$) is introduced by a turbulent component which depends on the subgrid-scale turbulent kinetic energy (TKE) described by Brinkop and Roeckner (1995). Thus, the vertical velocity is given by the sum of the grid mean vertical velocity ($\overline{w}$) and the turbulent contribution: $w = \overline{w} + 0.7\sqrt{TKE}$ (Lohmann and Kärcher, 2002).

## 2.2 Numerical representation of clouds

Convective cloud microphysics in EMAC is solely based on temperature and updraft strength and does not take into account the aerosol influence on cloud droplet and ice crystal formation. To simulate convective clouds, the CONVECT submodel includes multiple parameterizations which address the influence of the convective activity on the larger scale circulation by adding the detrained water vapour to the large-scale water vapour field. The detrained cloud condensate is used as a source term for the cloud condensate treated by the CLOUD submodel and is considered in the liquid or ice phase depending on its temperature (if temperature is lower than $-35°C$ the phase is ice, otherwise it is liquid). In this work, the scheme of Tiedtke (1989) with modifications by Nordeng (1994) has been used.

The CLOUD submodel describes physical and microphysical processes in large-scale stratiform clouds. It uses a double-moment cloud microphysics scheme for cloud droplets and ice crystals (Lohmann et al., 1999; Lohmann and Kärcher, 2002; Lohmann et al., 2007) and solves the prognostic equations for specific humidity, liquid cloud mixing ratio, ice cloud mixing ratio, cloud droplet number concentration (CDNC), and ICNC. Cloud droplet formation is computed by an advanced physically based parameterization (Kumar et al., 2011; Karydis et al., 2011) that merges two theories: the $\kappa$-Köhler theory (Petters and Kreidenweis, 2007), which governs the activation of soluble aerosols, and the Frenkel-Halsey-Hill adsorption activation theory (Kumar et al., 2009), which describes the droplet activation due to water adsorption onto insoluble aerosols (e.g. mineral dust). This parameterisation is applied to the aerosols that consist of an insoluble core with soluble coating, while soluble aerosols follow the $\kappa$-Köhler theory (Karydis et al., 2017). In the cirrus regime, ice crystals can form either via homogeneous nucleation, using the parameterization of Kärcher and Lohmann (2002b, KL02), or via homogeneous and heterogeneous nucleation using the parameterization of Barahona and Nenes (2009, BN09), which takes into account the competition for the available water vapour between the two ice nucleation mechanisms and among the pre-existing ice crystals (Bacer et al., 2018). In the mixed-phase regime, contact nucleation is simulated according to Lohmann and Diehl (2006, LD06). Immersion nucleation can be parameterized either via LD06 or via the empirical parameterization of Phillips et al. (2013, P13), which can also simulate deposition nucleation. Both LD06 and P13 are aerosol dependent. In this study, LD06 considers insoluble mineral dust for contact nucleation and soluble dust and black carbon for immersion nucleation, while P13 takes into account insoluble dust and black carbon, and glassy organics for immersion and deposition nucleation. (For a detailed comparison of the ice nucleation parameterizations BN09, KL02, LD06, and P13 we refer to Bacer et al. (2018).) Cloud cover is computed diagnostically with the scheme of Sundqvist et al. (1989), based on the grid-mean relative humidity. Other microphysical processes, like phase transitions, autoconversion, aggregation, accretion, evaporation of rain, melting of snow, are also taken into account by the CLOUD submodel.

| Tendency | Description | Temperature |
|----------|-------------|-------------|
| DETR | Convective detrainment | $T < -35°\text{C}$ |
| NCIR | Ice nucleation in the cirrus regime | $T < -35°\text{C}$ |
| FREE | Instantaneous freezing | $T < -35°\text{C}$ |
| NMIX | Ice nucleation in the mixed-phase regime | $-35°\text{C} < T < 0°\text{C}$ |
| SECP | Secondary ice production | $-8°\text{C} < T < -3°\text{C}$ |
| MELT | Melting | $T > 0°\text{C}$ |
| SELF | Self-collection | $T < 0°\text{C}$ |
| AGGR | Aggregation | $T < 0°\text{C}$ |
| ACCR | Accretion | $T < 0°\text{C}$ |

**Table 1.** ICNC tendencies of the microphysical processes defined in the CLOUD submodel. Sources of ICs are in the highest block, sinks of ICs are in the lowest block. The first column contains the abbreviations associated with each tendency; the second column describes the microphysical processes associated with each tendency; the third column specifies the temperature range in which the processes occur.

## 2.3 ICNC tendencies

### 2.3.1 Microphysical tendencies

According to Lohmann (2002) and Roeckner et al. (2004), the evolution of ICNC (i.e. rate or *tendency* of ICNC) is described by the following prognostic equation:

$$\frac{\partial \text{ICNC}}{\partial t} = R_{transp} + R_{sedi} + R_{ncir} + R_{nmix} + R_{secp} - (R_{self} + R_{aggr} + R_{accr} + R_{melt} + R_{subl}) \tag{1}$$

where the $R$-terms (in $\text{m}^{-3}\text{s}^{-1}$) are the ICNC tendencies due to specific physical or microphysical processes: advective, turbulent, and convective transport ($R_{transp}$), sedimentation ($R_{sedi}$), ice nucleation in the cirrus regime ($R_{ncir}$), ice nucleation in the mixed-phase regime ($R_{nmix}$), secondary ice production ($R_{secp}$), self-collection ($R_{self}$), aggregation ($R_{aggr}$), accretion ($R_{accr}$), melting ($R_{melt}$), and sublimation ($R_{subl}$) of ice crystals. Transport as well as sedimentation of ICs are computed for the grid-box volume ($\overline{\text{ICNC}}$), while the other terms are in-cloud processes (ICNC$_{\text{in-cloud}}$). The latter ones are related to the grid-mean values via the fractional cloud cover ($f_C$): ICNC$_{\text{in-cloud}} = \overline{\text{ICNC}}/f_C$. Among the processes in equation (1), advective, turbulent, and convective transport and sedimentation (which vertically redistributes the ICs and is formally treated like vertical advection) are physical processes solved by the model, while all others are microphysical processes computed with different parameterizations.

In this work, we decompose the microphysical sources and sinks of ICs in the CLOUD submodel (Table 1), i.e. all $R$-terms except $R_{sedi}$ and $R_{transp}$. It must be mentioned that sublimation of falling ICs that encounter an ice subsaturated region has not been analysed in this work as it is not explicitly treated.

**Sources of ice crystals.** The number of new ICs originating from convective detrainment (DETR) is estimated from the detrained cloud condensate in the ice phase (i.e. when temperature is lower than $-35°\text{C}$, see Subsection 2.2) by assuming a temperature dependent IC radius. DETR is included in the transport term of equation (1) (Roeckner et al., 2004), but it will

be studied here as an independent source of ICs defined within the CLOUD submodel. As described in Subsection 2.2, ice crystal formation in the cirrus regime (NCIR) is simulated via the ice nucleation parameterizations BN09 or KL02. Moreover, supercooled cloud droplets freeze instantaneously (FREE), i.e. they glaciate in one time step, when they are transported to regions where temperature is below $-35°C$ (like in Levkov et al., 1992). In the mixed-phase regime, the number of new ICs formed via heterogeneous nucleation (NMIX) is the sum of the ice crystals originated from contact, immersion/condensation, and deposition nucleation modes, i.e. the results of the heterogeneous nucleation parameterizations LD06 and/or P13 applied in this regime. Secondary ice production (SECP) occurs via the Hallet-Mossop process between $-3°C$ and $-8°C$ as described in Levkov et al. (1992). NCIR represents in-situ cirrus clouds, those forming at temperatures colder than $-35°C$ via heterogeneous or homogeneous ice nucleation of solution droplets. FREE represents liquid-origin cirrus, whose ICs are generated by the advection of already-formed water cloud droplets below $-35°C$; this process is particularly active in regions with mesoscale convective activity and warm conveyor belts (Krämer et al., 2016). Also immersion and contact nucleation contribute to form liquid-origin cirrus (Wernli et al., 2016), but they are considered in NMIX here.

**Sinks of ice crystals.** In general, self-collection (SELF), aggregation (AGGR), and accretion (ACCR) of ice crystals are based on the approach described in Lin et al. (1983). More precisely, collection efficiency of aggregation depends on snow crystal size according to Lohmann (2004), collection efficiency of accretion is temperature dependent following Levkov et al. (1992), while collection efficiency of self-collection is constant like in Levkov et al. (1992). It is assumed that ice crystals melt instantaneously (MELT) as soon as temperature is above $0°C$ and are converted into cloud droplets.

### 2.3.2 Numerical tendencies

The CLOUD submodel also includes ICNC tendencies that impose specific values when particular conditions are satisfied. For example, if ICNC exceeds an upper threshold of $ICNC_{max} = 10^7$ m$^{-3}$, the ICNC value is replaced by $ICNC_{max}$, forcing a sudden decrease of ICNC within one time step. These correction terms do not have a physical meaning and we will refer to them as *numerical tendencies* (Table 2). Their role has rarely been addressed in the literature.

### 3 Setup of simulations

The simulations in this study have been performed at T42L31ECMWF resolution, which corresponds to a spherical truncation of T42 (i.e. quadratic Gaussian grid of approximately $2.8° \times 2.8°$, in latitude and longitude) and 31 vertical hybrid pressure levels up to 10 hPa (about 25 km). The model time step is 20 minutes, and the model results are stored with a frequency of 5 hours. The simulations run for 6 years: the first year has been considered spin-up time, while the next five years have been used for the analysis. Two periods are taken into account: the years 2000-2005 to represent present-day conditions and the years 2080-2085 to represent a global warming scenario. The simulations are forced by prescribed sea surface temperatures (SSTs) and sea-ice concentrations (SICs). SSTs and SICs are provided by the Hadley Centre Global Environment Model version 2 – Earth System (HadGEM2-ES) Model (Collins et al., 2011): the historical simulation with HadGEM2-ES is used for the present period, while the RCP6.0 simulation is considered for the future (like in the RC2-oce-01 simulation of the ESCiMo project

| Tendency | Description | Mean |
|---|---|---|
| minmax0 | The minimal value allowed for ICNC is imposed ($ICNC_{background} = 10^{-12}$ m$^{-3}$) | $10^{-16}$ |
| minmax1 | The maximal ICNC correction term is imposed ($ICNC_{max} = 10^7$ m$^{-3}$) | $-10^1$ |
| minmax2 | The minimal ICNC correction term is imposed ($ICNC_{min} = 10$ m$^{-3}$) | $10^{-4}$ |
| minmax3 | ICNC is equal to concentrations of the new ICs produced in the cirrus regime *(1)* | $-10^{-2}$ |
| minmax4 | $ICNC_{min}$ is imposed | $10^{-2}$ |
| minmax5 | ICNC is equal to $ICNC_{background}$ *(2)* | $-10^{-1}$ |
| minmax6 | $ICNC_{background}$ is guaranteed | 0 |
| minmax7 | Removal processes can decrease ICNC at maximum by the same value ICNC | 0 |
| minmax8 | $ICNC_{background}$ is guaranteed | $10^{-24}$ |
| minmax9 | $ICNC_{min}$ is imposed | $10^{-3}$ |

**Table 2.** Numerical ICNC tendencies defined in the CLOUD submodel. The third column shows the order of magnitude (in m$^{-3}$s$^{-1}$) of the global means computed with the REF simulation. *(1)* when the condition (cloud cover > 0 & cloud ice > $10^{-12}$ kg kg$^{-1}$ & ICNC < $ICNC_{min}$) is true; *(2)* when the condition (cloud cover > 0 & cloud ice > $10^{-12}$ kg kg$^{-1}$) is false.

| Simulation | Ice nucleation scheme | | Years |
|---|---|---|---|
| | *Cirrus regime* | *Mixed-phase regime* | |
| REF | BN09 | cnt: LD06 ; imm&dep: P13 | 5 years (around 2000) |
| PRES | KL02 | cnt: LD06 ; imm: LD06 | 5 years (around 2000) |
| FUT | BN09 | cnt: LD06 ; imm&dep: P13 | 5 years (around 2080) |
| NOicncmax | BN09 | cnt: LD06 ; imm&dep: P13 | 1 year (around 2000) |
| NOfree | BN09 | cnt: LD06 ; imm&dep: P13 | 1 year (around 2000) |

**Table 3.** Simulations carried out and analysed in this study. The abbreviations "cnt" and "imm&dep" stand for contact nucleation and immersion & deposition nucleation, respectively.

described in Jöckel et al., 2016). Aerosols are emitted offline using monthly emission files based on the AEROCOM data set, such as for mineral dust, secondary organic aerosol, and sea salt (like in Pozzer et al., 2012), or a combination of the ACCMIP (Lamarque et al., 2010) and RCP 6.0 scenario (Fujino et al., 2006), such as for black carbon and organic carbon with biomass burning and anthropogenic origins.

The simulations carried out in this study are one reference run and two sensitivity case studies (Table 3). The reference run (REF) simulates recent conditions and applies the ice nucleation parameterizations BN09 and P13 in the cirrus regime and mixed-phase regime, respectively (like in Bacer et al., 2018). REF will be analysed in order to quantify the rates of ice microphysical processes in cold clouds and define their relative importance. Another simulation (PRES) refers to the same period but uses different ice nucleation schemes in order to understand the effect of parameterization choice. In particular, the PRES simulation uses KL02 in the cirrus regime and LD06 in the mixed-phase regime. Finally, the simulation representing the future period (FUT) has been run with the same model setup as REF but with the RCP6.0 emission scenario. The comparison

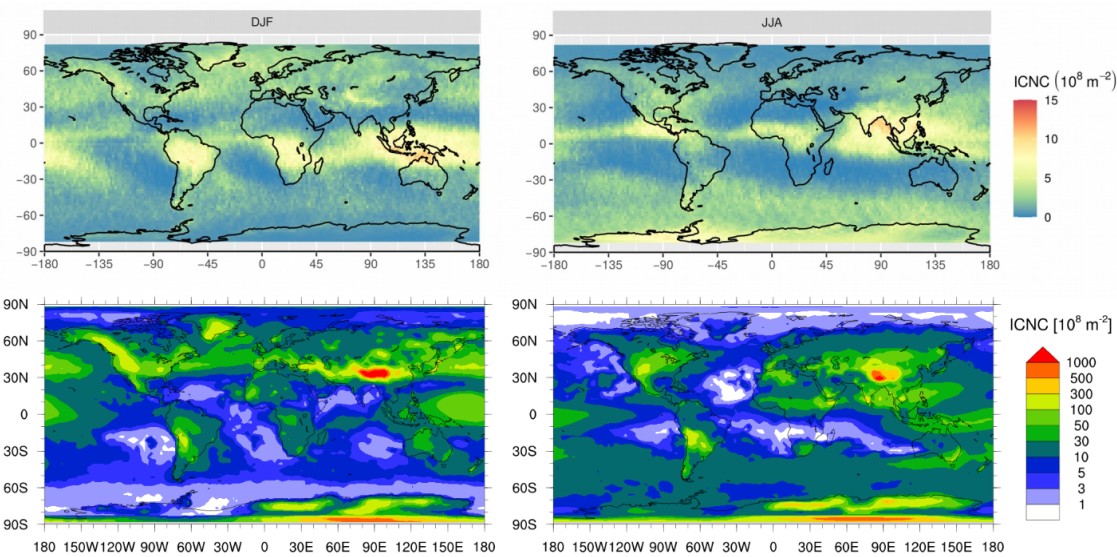

**Figure 1.** Mean spatial distribution of vertically integrated ICNC burden for the DJF and JJA seasons. (*Top*) In-cloud ICNC burden retrieved by DARDAR-Nice (2006–2017) averaged in a 2°× 2°grid. (*Bottom*) In-cloud ICNC burden computed by EMAC (REF 5-hour output greater than zero were considered in the average).

between FUT and REF will allow us to estimate the changes in cold cloud microphysical processes under a global warming
scenario.

Additionally, two 2-year test simulations (2000 for spin-up time and 2001 for the analysis) have been run (Table 3). Both tests use the same setup as REF. In NOicncmax, the condition that ICNC must be lower than $ICNC_{max}$ at each model time step (i.e. the numerical tendency minmax1) is dropped, allowing us to investigate the impact of the largest numerical tendency (Table 2). In NOfree, supercooled cloud droplets can remain liquid also at temperatures lower than $-35°$C in order to understand the
influence of the FREE tendency.

## 4   Model results and evaluation of ICNC

In this section, the ICNC obtained with the EMAC model is investigated and evaluated through comparisons to satellite ICNC retrievals by DARDAR (lidDar-raDAR)-Nice (Gryspeerdt et al., 2018; Sourdeval et al., 2018). This satellite product uses the sensitivity contained in combined space-borne lidar-radar measurements in order to constrain the parameters of the particle size distribution (PSD) then used to infer the ICNC by direct integration from a particle size of 5 $\mu$m. DARDAR-Nice retrievals are
provided at vertical and horizontal resolutions of 60 m and 1.4 km, respectively. This data set has been thoroughly evaluated against a large variety of in-situ measurements (Sourdeval et al., 2018; Krämer et al., 2020), to find an overall agreement within a factor of two at cirrus temperatures. However, it should be noted that an overestimation of ICNC at warmer temperatures

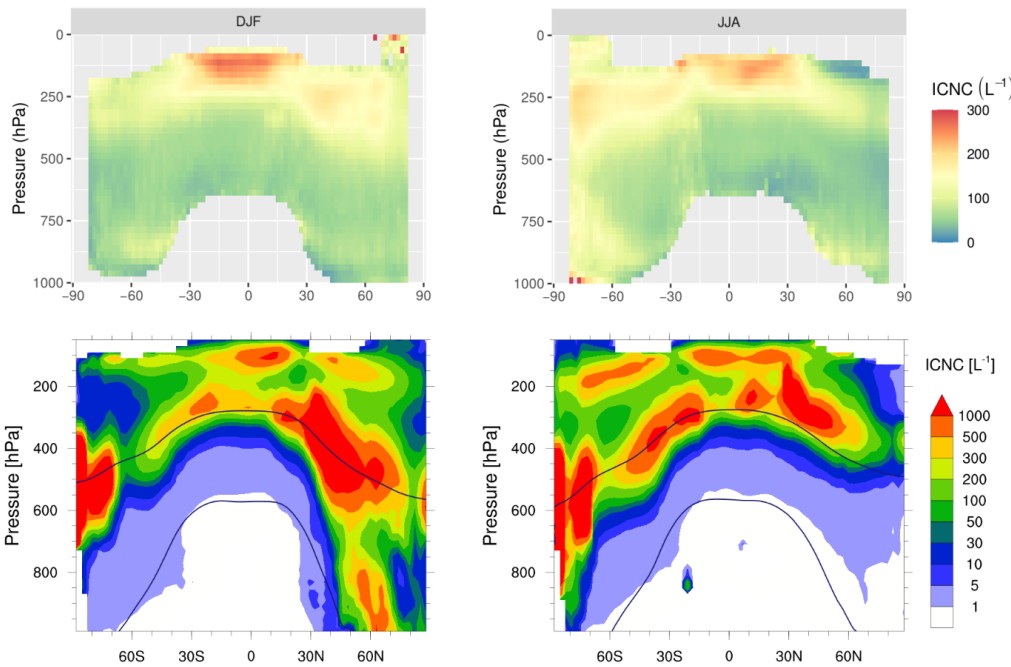

**Figure 2.** Zonal means of in-cloud ICNC for the DJF and JJA seasons by DARDAR-Nice (*top*) and EMAC (REF, *bottom*); the isotherms at $0°C$ and $-35°C$ are seasonal means.

is possible due to the misrepresentation of the PSD bi-modality by the satellite retrieval method and of optical properties of
mixed-phase clouds.

Figure 1 shows the spatial distributions of the ICNC burden during winter (DJF) and summer (JJA) seasons for both a 10-year climatology of DARDAR-Nice retrievals and the model results of the REF simulation. The satellite products present high ICNC values mainly in deep convective regions as well as in mid-latitudes during winter months, possibly due to increased ice nucleation rates associated with high wind velocities. Such features are in most parts also observed in the patterns of the
model ICNC burden distribution, which is in good overall agreement with the satellite retrievals. However, absolute values differ by about an order of magnitude, with ICNC burdens up to about $10^9$ m$^{-2}$ in DARDAR-Nice and $10^{10}$ m$^{-2}$ in EMAC in most of these two regions. A larger discrepancy can be seen over Antarctica, where the model overestimates ICNC probably due to very low temperatures (lower than $-35°C$ most of the year) and high supersaturation levels. Even higher values, up to $10^{11}$ m$^{-2}$, are simulated by EMAC in mountainous regions. ICNCs of the same order of magnitude can be found in other
modeling studies (e.g. Kuebbeler et al., 2014; Gasparini and Lohmann, 2016; Bacer et al., 2018). Although increases of ICNC around steep orography are noticed in the satellite products, and are consistent with strong homogeneous freezing in the strong uplifts associated with mid-latitude jets during winters, they mainly occur right between the homogeneous nucleation threshold and $-60°C$ (Sourdeval et al., 2018), where ICNC locally reaches up to $300$ L$^{-1}$ (nearly three times the surrounding values). Therefore, these features do not strongly appear in the ICNC burden nor in the corresponding zonal ICNC profiles

shown in Figure 2 (top). These profiles exhibit ICNC values that are consistent with the aforementioned observations, i.e. high ICNC values (up to 300 L$^{-1}$) in the tropics and in the mid-latitudes (up to 150 L$^{-1}$). Sharp increases of ICNC values (from about 50 to above 100 L$^{-1}$) are also noted in the vertical profiles between 500 and 300 hPa, according to the activation of homogeneous nucleation. These features are consistent with what is modeled in EMAC (Figure 2, bottom), both in terms of patterns and absolute values. Nevertheless, higher ICNC values, up to 1000 L$^{-1}$, tend to occur at lower altitude in the

troposphere, seemingly related to orographic features. While uncertainties remain on the absolute ICNC by DARDAR-Nice, it should be noted that such high values are only rarely reported from in-situ measurements (Krämer et al., 2016, 2020), therefore it is likely that EMAC overestimates ICNC. Interestingly, the ICNC zonal mean computed with the NOfree simulation (not shown) is closer to the observations with respect to the REF simulation, suggesting that the FREE tendency contributes to the overestimation of ICNC.

## 5    ICNC tendency results

### 5.1    Global statistics

In this section, we analyse the role of each tendency in terms of extent (Table 4) and relative contribution (Figure 3) at the global scale. In all simulations, the largest source of ICs is the instantaneous freezing, whose mean tendency is of the order of $10^2$ m$^{-3}$s$^{-1}$ with a relative contribution of about 50%. FREE is followed by convective detrainment and homogeneous and

heterogeneous ice nucleation in the cirrus regime. In mixed-phase clouds, the largest IC source is heterogeneous nucleation, followed by secondary ice production; they are of the order of $10^{-2}$ m$^{-3}$s$^{-1}$, and their relative contribution is less than 0.1%. Globally, the hierarchy of IC sources in the REF simulation is FREE > DETR > NCIR > NMIX > SECP (Table 4). Our results are in agreement with the recent study of Muench and Lohmann (2020), who also found that homogeneous freezing and convective detrainment are the dominant sources of ICs. Aggregation is the major physical removal process of ICs in all

simulations, of the order of 10 m$^{-3}$s$^{-1}$, with about double the rate of accretion. Self-collection and melting are much less efficient sinks, on average two to four orders of magnitude lower than AGGR, respectively, with a relative contribution smaller than 0.1%. Hence, the hierarchy of IC sinks in REF is AGGR > ACCR > SELF > MELT (Table 4).

At this point, the important role of the numerical tendencies must be stressed. While most numerical tendencies have contributions to ICNC smaller than any of the microphysical tendencies (Tables 2 and 4), a few have non-negligible contributions

(e.g. minmax1,3,4,5). As a result, the sum of all negative numerical tendencies (MINMAX-) is higher than AGGR, for example, contributing more than 30% to IC removal, relative to only 10% from AGGR. These correction terms are not often analysed, but we highlight their importance here. Ice microphysics parameterizations may get the right answer for the wrong reason because of these numerical artifacts.

We can illustrate the impact of these numerical tendencies by examining the test simulation NOicncmax. The imposition

of ICNC$_{max}$ (minmax1) is the dominant negative numerical tendency (Table 2). Without this condition, MINMAX- decreases by an order of magnitude, and ACCR and AGGR become the dominant sink terms (Figure 3). Moreover, while there is a quite balanced division between IC sources and sinks for the other simulations, the source terms dominate in NOicncmax at

| Tendency | REF | | PRES | | FUT | | NOicncmax | | NOfree | |
|---|---|---|---|---|---|---|---|---|---|---|
| | Mean | StDev | Mean | StDev | Mean | StDev | Mean | StDev | Mean | StDev |
| DETR | 1.8e+00 | 6.2e+01 | 1.5e+00 | 5.6e+01 | 1.7e+00 | 5.6e+01 | 1.7e+00 | 5.5e+01 | 2.4e+00 | 8.7e+01 |
| NCIR | 6.0e-01 | 9.9e+00 | 4.7e+01 | 2.1e+03 | 5.0e-01 | 7.2e+00 | 4.9e-01 | 8.6e+00 | 4.5e-01 | 8.2e+00 |
| FREE | 1.1e+02 | 3.6e+03 | 9.2e+01 | 3.3e+03 | 7.8e+01 | 3.0e+03 | 1.1e+02 | 3.5e+03 | / | / |
| NMIX | 5.6e-02 | 2.6e+00 | 2.4e-01 | 2.1e+01 | 3.9e-02 | 2.0e+00 | 4.9e-02 | 2.2e+00 | 6.3e-02 | 3.1e+00 |
| SECP | 1.7e-02 | 3.4e-01 | 1.6e-02 | 3.3e-01 | 1.5e-02 | 3.1e-01 | 1.6e-02 | 3.3e-01 | 1.7e-02 | 3.4e-01 |
| MELT | -1.5e-03 | 9.6e-01 | -1.4e-03 | 9.9e-01 | -1.4e-03 | 1.1e+00 | -2.2e-03 | 2.4e+00 | -6.9e-04 | 5.6e-02 |
| AGGR | -1.7e+01 | 4.9e+02 | -1.6e+01 | 4.3e+02 | -1.3e+01 | 4.3e+02 | -4.0e+01 | 2.0e+03 | -9.9e-01 | 1.8e+01 |
| ACCR | -9.1e+00 | 3.3e+02 | -8.2e+00 | 2.9e+02 | -6.8e+00 | 2.6e+02 | -2.3e+01 | 1.1e+03 | -3.9e-01 | 6.9e+00 |
| SELF | -1.1e-01 | 3.6e+00 | -1.0e-01 | 3.1e+00 | -8.3e-02 | 3.1e+00 | -3.0e-01 | 1.7e+01 | -3.6e-03 | 6.1e-02 |
| MINMAX+ | 4.9e-02 | | 3.9e-03 | | 4.6e-02 | | 6.5e-02 | | 2.7e-02 | |
| MINMAX- | -6.7e+01 | | -9.7e+01 | | -4.6e+01 | | -3.5e+00 | | 1.1e+00 | |
| ICNC | 2.5e+02 | 7.3e+03 | 2.6e+02 | 6.5e+03 | 1.8e+02 | 5.9e+03 | 7.8e+02 | 3.1e+04 | 1.5e+01 | 1.3e+02 |
| Hierarchy (REF) | Sources: FREE > DETR > NCIR > NMIX > MINMAX+ > SECP<br>Sinks: MINMAX- > AGGR > ACCR > SELF > MELT | | | | | | | | | |

**Table 4.** Statistics computed by using 5-hourly output of the 5-year simulations REF, PRES, and FUT and the 1-year simulations NOicncmax and NOfree. Global means and standard deviations are in $\mathrm{m}^{-3}\mathrm{s}^{-1}$ for the tendencies and in $\mathrm{L}^{-1}$ for grid-averaged ICNC. MINMAX+ and MINMAX- are the sum of the means of positive and negative numerical tendencies, respectively (according to Table 2). The last two rows summarise the hierarchy of the ICNC tendencies in REF.

60%. We have not considered the transport and sedimentation tendencies here and so cannot determine whether the clouds can realistically dissipate in the absence of the minmax1 tendency. However, we can emphasise the impact of this numerical tendency as the global mean ICNC in the NOicncmax simulation is three times larger than that in the REF simulation (Table 4). Therefore, an enforced ICNC upper bound of $10^7$ $\mathrm{m}^{-3}$ significantly dampens the ICNC produced globally (Figure S1 in the Supplement).

We also investigated the case in which the dominant source, FREE, does not take place. The results of the test simulation NOfree show that the ICNC tendencies remain of the same magnitude (Table 4). The suppression of instantaneous freezing does allow detrainment to become the leading source of ICs (Figure 3). ICNC also strongly decreases in the middle and lower troposphere (Figure S1), while global mean ICNC drops by an order of magnitude with respect to the REF simulation (Table 4). In contrast, CDNC increases by 10% on average (not shown), as cloud droplets that would otherwise transform into ICs in REF remain in the liquid phase in NOfree.

Finally, for each microphysical process, we computed the occurrence of the tendency values greater than zero (Figure 4). We find that all distributions are highly asymmetric and, in particular, left-skewed. Only MELT shows a bell-shaped distribution; but even in this case, the median is lower than the mean suggesting a tail to the left of the distribution. A few processes are characterised by multimodal distributions; for example, the distribution of DETR is bimodal, while the distributions of SELF, AGGR, and ACCR are trimodal.

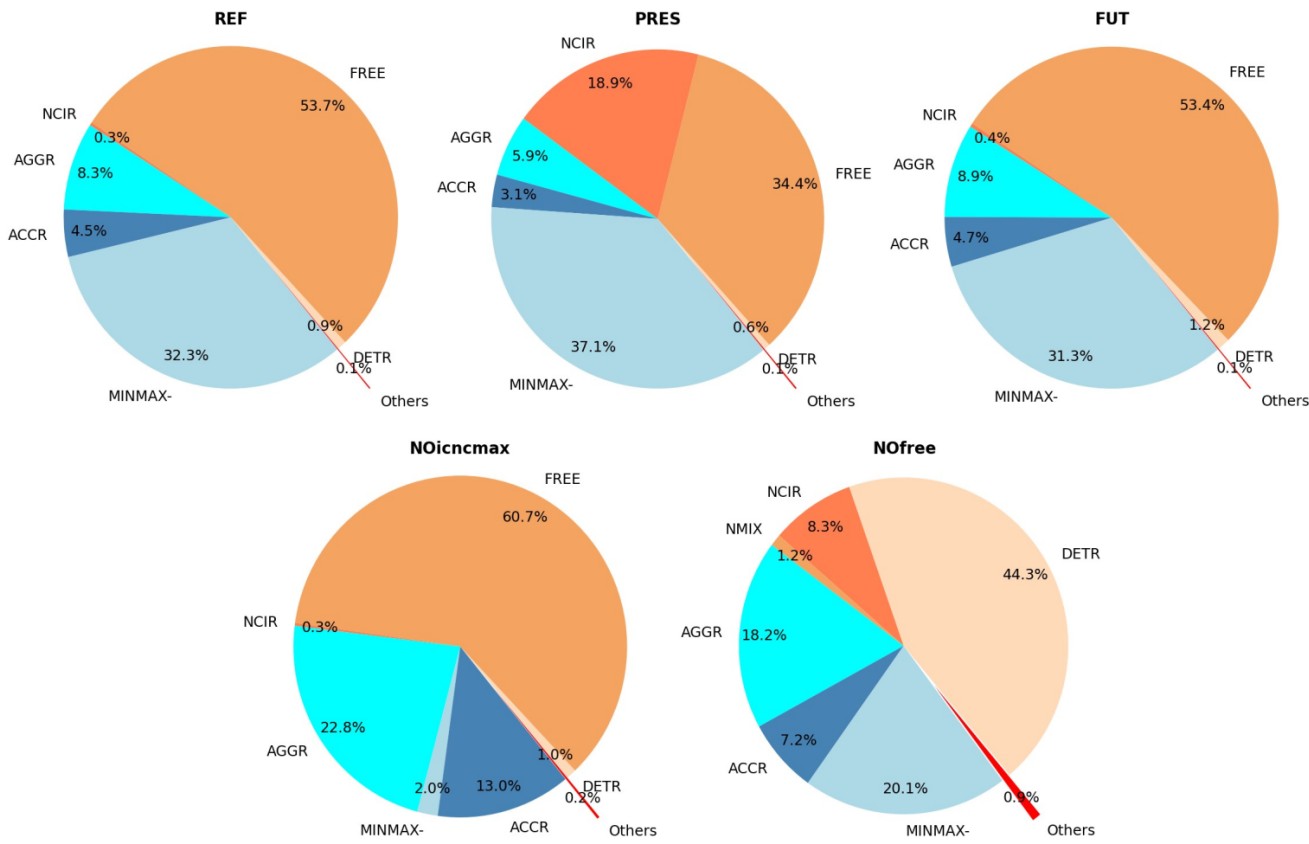

**Figure 3.** Relative contributions of the mean tendencies in Table 4. The sector "Others" of the pie charts includes NMIX, SECP, MELT, SELF, MINMAX+ (apart in NOfree, where NMIX is represented independently). Warm tones of colors indicate sources of ICs, while cold tones of colors indicate sinks of ICs.

## 5.2 Spatial distributions

The global distributions of the vertically integrated tendencies for the REF simulation are shown in Figure 5. Both DETR and NCIR are higher over regions that experience strong convective activity, e.g. the Intertropical Convergence Zone (ITCZ) and the Tropical Warm Pool (TWP). DETR is higher over land than over ocean because the land-ocean differences in the thermodynamic profiles below the freezing level produce stronger updrafts over land (Del Genio et al., 2007). DETR and NCIR tend to be smaller off the west coasts of South America, Africa, and Australia where SSTs are colder and stratocumulus decks dominate. FREE mostly occurs in extratropical regions, where warm conveyor belts can form, and over continents. In particular, FREE shows high values over mountainous regions, where liquid cloud droplets are efficiently transported by strong updrafts up to levels where the temperature is lower than $-35°$C and freeze, and over Antarctica, where the temperature is lower than the freezing threshold for most of the year. The high values of FREE could be responsible for the ICNC overestimation (as

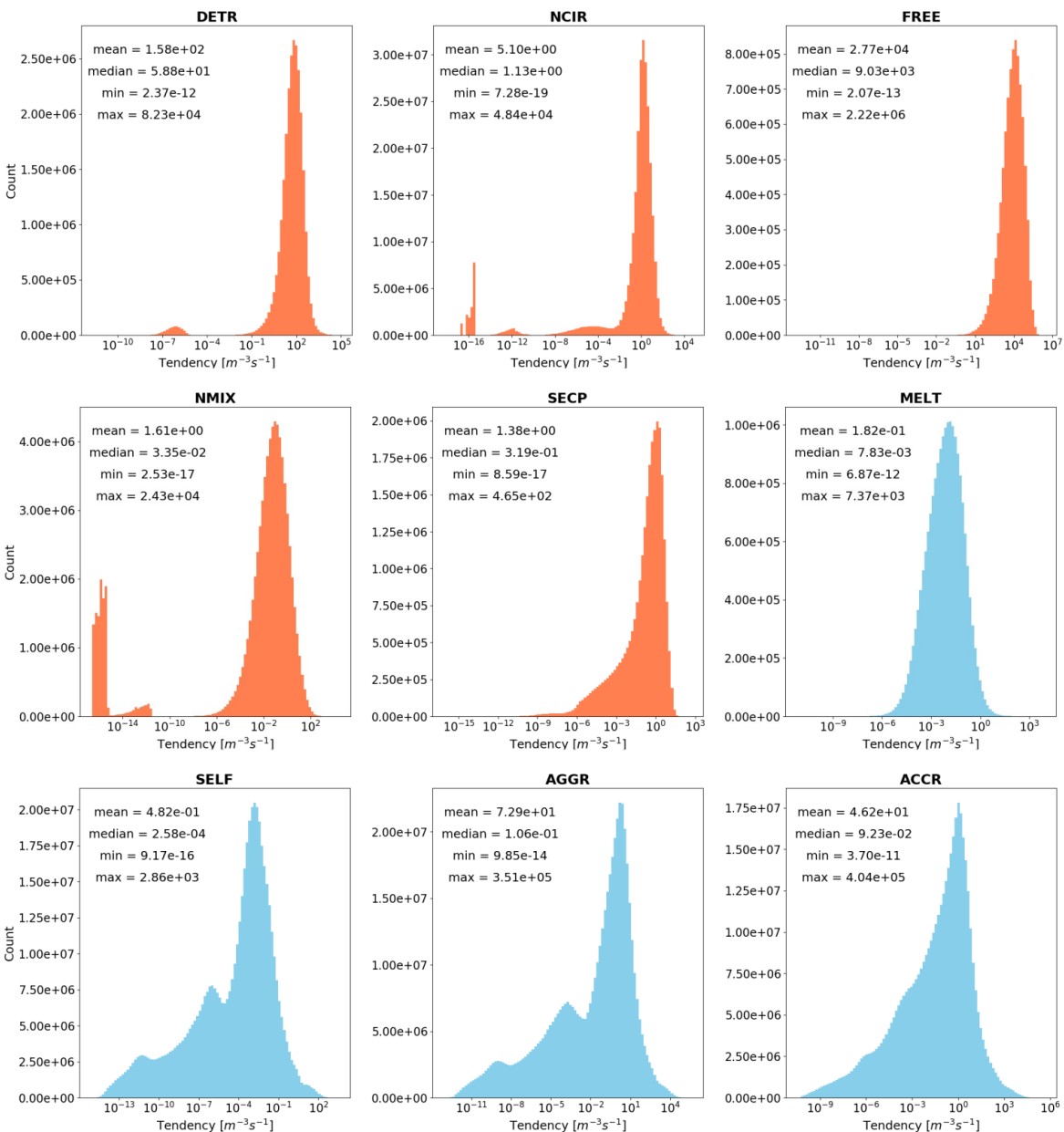

**Figure 4.** Occurrence and statistics of ICNC tendencies (REF). The bar charts are computed with 5-hour output data distributed in 100 logarithmic bins. For each tendency, only values grater than zero have been considered in the analysis (absolute values are used for the sinks). The vertical axis shows the occurrence in linear scale, the horizontal axis shows the tendency values in logarithmic scale. Warm tones of colors indicate sources of ICs, while cold tones of colors indicate sinks of ICs.

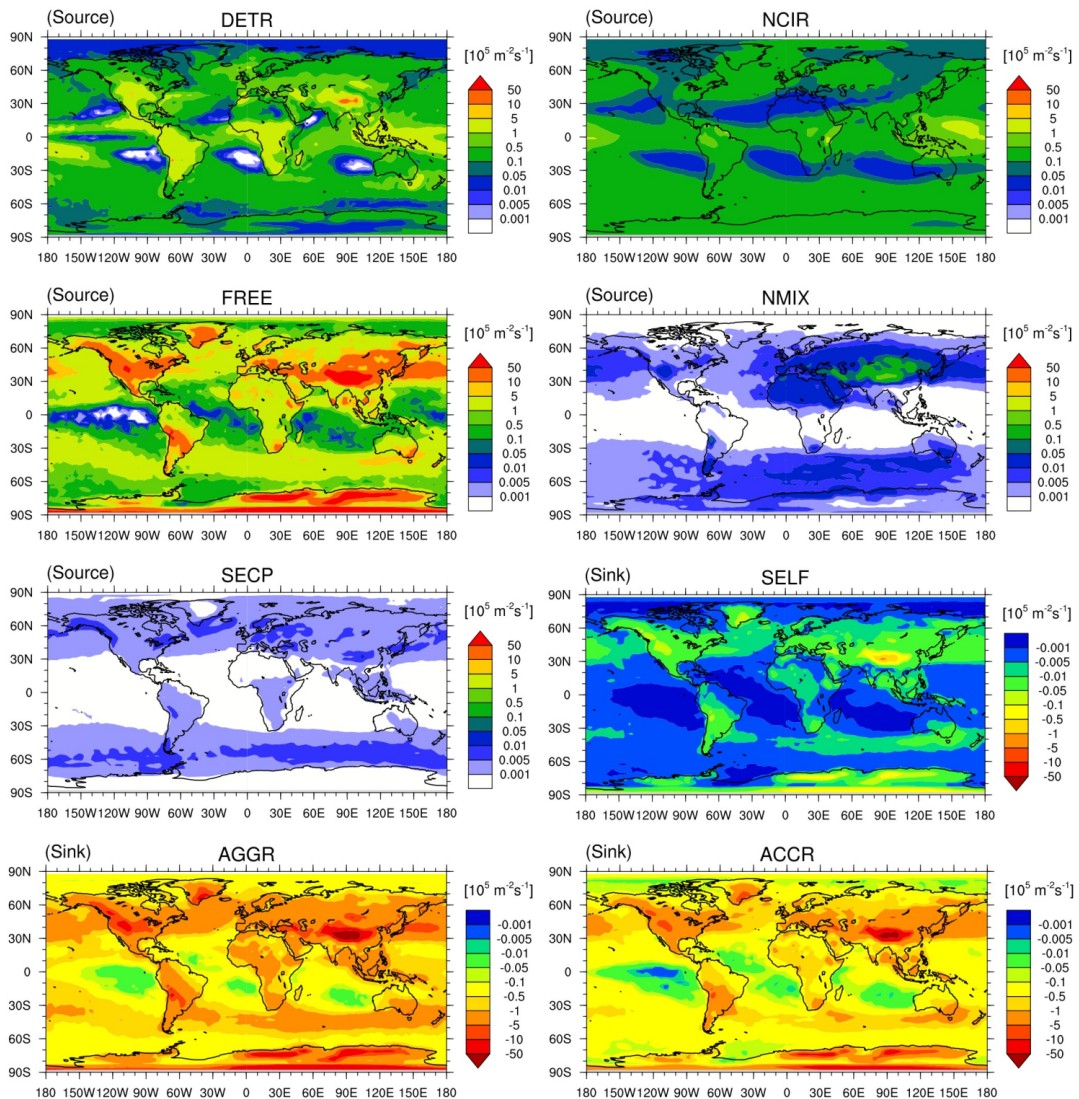

**Figure 5.** Annual means of the vertically integrated tendencies (in $10^5$ m$^{-2}$s$^{-1}$) for sources and sinks of ICs in cold clouds (REF).

mentioned in Section 4). Since FREE contributions are high but localised, their annual mean is larger than DETR and NCIR

305  while the FREE annual median is negligible (Subsection 5.4). NMIX is influenced by the orography and the abundance of the

INPs responsible for heterogeneous nucleation in the P13 scheme: the largest tendencies occur over the Rocky Mountains, the

Andes, the Himalayas, and over and downwind of large deserts (e.g. the Saharan region and the Arabian peninsula). NMIX is

also large over Asia due to high emissions of black carbon and dust from the Gobi Desert. All IC sinks show similar patterns

globally: they are higher over land and influenced by orography. They are also high throughout the mid-latitudes and over

310  Antarctica, following the vertically integrated ICNC pattern (Figure 1).

## 5.3 Zonal means

We next explore the zonally averaged profiles of IC sources and sinks in the REF simulation (Figures 6 and 7). We clearly see that ice nucleation in the cirrus regime (NCIR) is the dominant source of ICs at pressures lower than 200 hPa between the tropics and lower than 350 hPa at high latitudes. NCIR presents a maximum in the tropical upper troposphere, coincident with the maximum of ICNC (Figure S1 in the Supplement), where temperature is very low ($T < -80°C$ on average) and ice supersaturation is high ($s_i > 36\%$ on average, not shown). NCIR is slightly higher in the Southern Hemisphere (SH) than in the Northern Hemisphere (NH), where heterogeneous nucleation occurs more frequently and could suppress homogeneous nucleation. DETR contributes to produce ICs at $T < -35°C$ (i.e. in the cirrus regime) especially between the mid-latitudes (50°N and 50°S), as illustrated also in Figure 5. By definition, detrained cloud condensate is in the ice phase when $T < -35°C$ (see Subsection 2.3.1), however, Coopman et al. (2020) have recently found that glaciation of isolated convective clouds over Europe usually occurs at higher temperature ($-22°C$). Hence, the temperature threshold for the cloud thermodynamic phase transition in the CLOUD submodel could be too low and contribute to an underestimation of ICNC in the mixed-phase regime with respect to observations (as discussed in Section 4). FREE is the largest source of ICs close to the transition from the cirrus to mixed-phase regime and especially outside the tropics. In mixed-phase clouds, NMIX dominates in the mid-latitudes, with values higher in the NH than in the SH because of higher INP and cloud droplet concentrations (e.g. Hoose et al., 2010; Liu et al., 2012; Karydis et al., 2017). While NMIX affects the whole mixed-phase regime, SECP is active at lower altitudes, as the Hallett-Mossop process occurs at $-8°C < T < -3°C$. In general, the zonal means of IC sources, but also their global distributions, are in agreement with the results of Muench and Lohmann (2020).

Vertical distributions of AGGR and ACCR are qualitatively similar (Figure 7). All sink processes except melting show higher values along the transition zone between the two cloud regimes, in particular in the NH and over the Antarctica where ICNCs are higher (Figure 1). AGGR extends to lower altitudes in the NH than in the SH.

It must be stressed that the IC sources and sinks of Figures 6 and 7 cannot be expected to balance for the following reasons. First, the tendencies of physical processes are not computed in this study, i.e. transport due to advection, turbulence, and convection and sedimentation ($R_{transp}$ and $R_{sedi}$ in equation (1), respectively). In particular, $R_{transp}$ is not computed in the CLOUD submodel but derives from various submodels in EMAC, e.g. CVTRANS (Tost et al., 2010) and E5VDIFF (Roeckner et al., 2004). Second, sublimation is a missing sink in this study. Finally, numerical tendencies also affect ICNC at each model time step and play a significant role in the ICNC budget (as discussed in Subsection 5.1).

## 5.4 Regional results

The ICNC tendencies are further analysed at the regional scale, considering areas over the Sahara, Amazon, Central Europe, North Atlantic Ocean, and Southern Indian Ocean (Figure S2 in the Supplement). For each region, the medians of the tendencies are computed in bins of 25 hPa and only in grid-boxes where ICNC $> 1$ L$^{-1}$ (Figure 8). The tendencies of different regions must then be compared along with the associated ICNC profiles, as a different number of grid-boxes is used for the statistics at the same vertical level. The lower the latitude, the higher the altitude associated with the peak in the ICNC profiles, as expected.

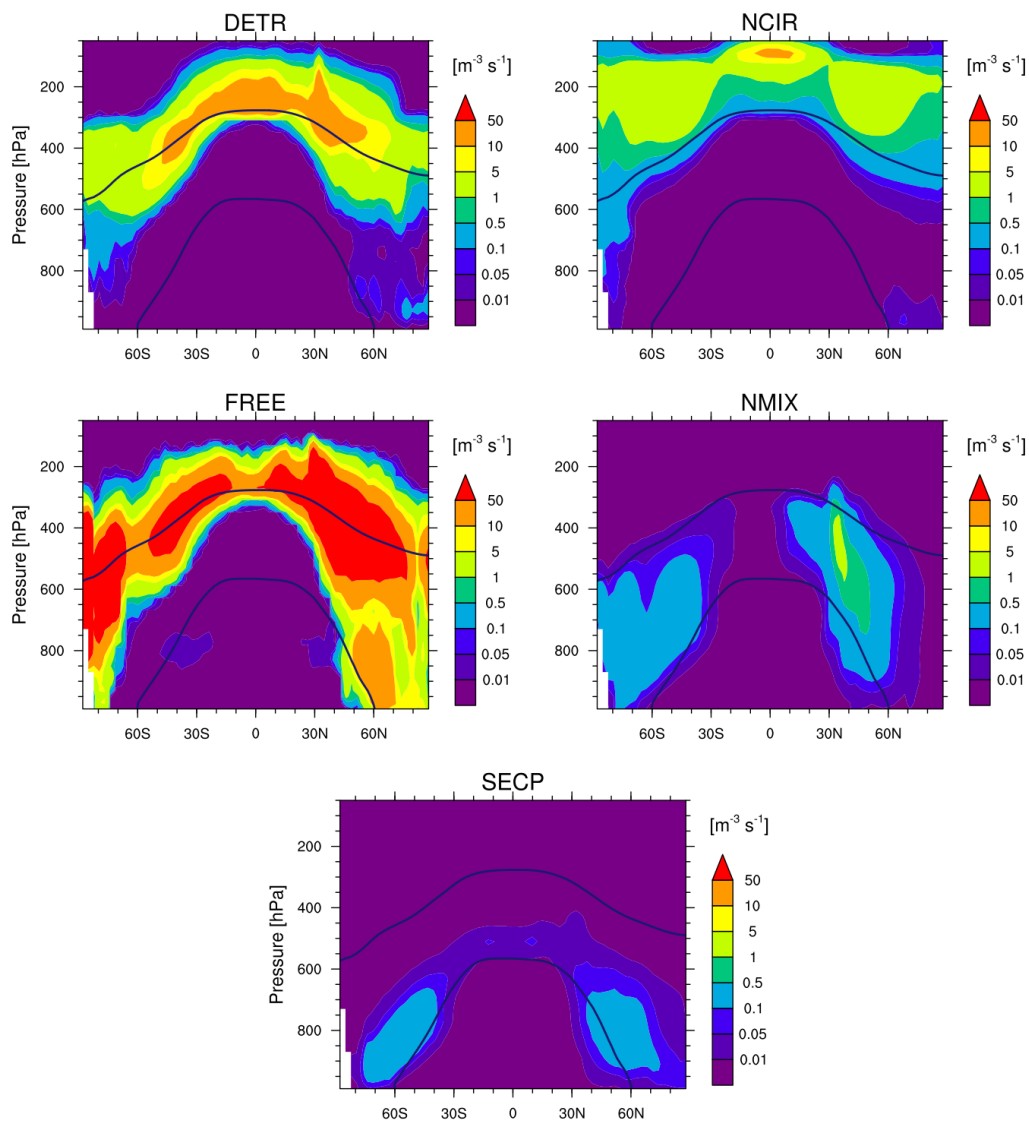

**Figure 6.** Annual zonal means of the tendencies associated to the IC sources in cold clouds (REF). The isotherms at $0°C$ and $-35°C$ are annual means.

Relatively colder surface temperatures over Europe mean both that the European ICNC maximum occurs at a lower pressure
345   level and that non-zero tendencies extend down to the surface.

In all regions, the sinks look similar: AGGR is a stronger removal process than ACCR, and its maximum is at higher altitudes than ACCR. The sources are more regionally variable. In the middle and lower troposphere over Europe, the Amazon and the maritime regions, ICs are generated by secondary ice production. The great relevance of secondary ice in regions with modest updrafts and aerosol loadings has been reported in several studies (e.g. Sullivan et al., 2016; Field et al., 2017). In contrast, over

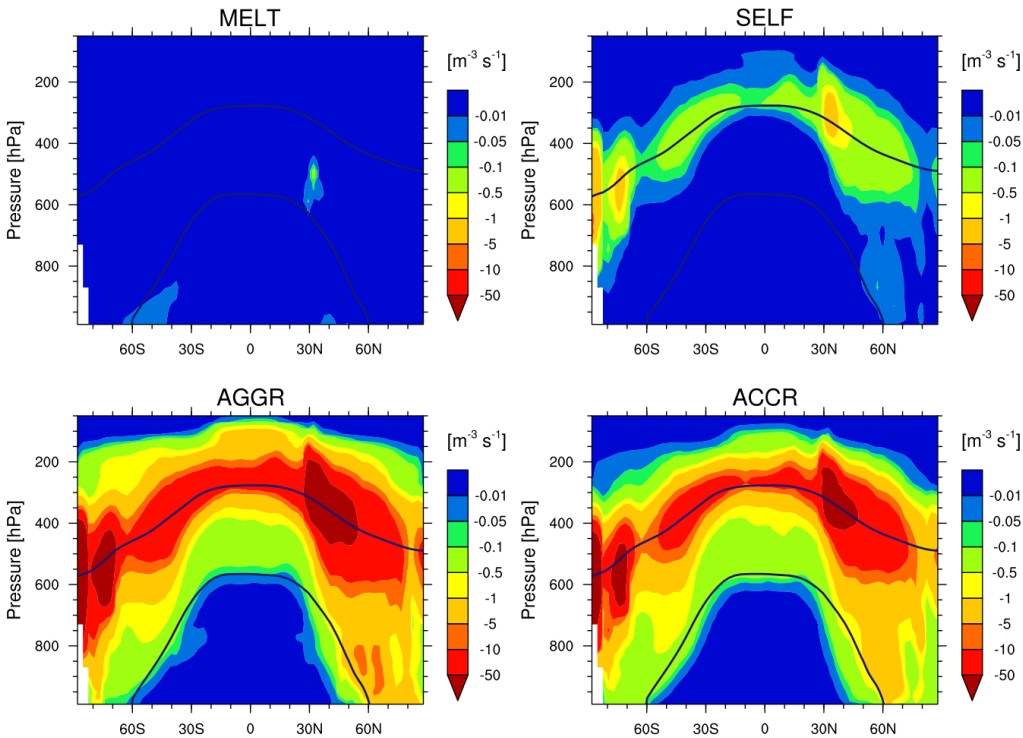

**Figure 7.** Annual zonal means of the tendencies associated to the IC sinks in cold clouds (REF). The isotherms at $0°C$ and $-35°C$ are annual means.

the Sahara, NMIX is the dominant IC source, given the large mineral dust loading. The regional means of the ICNC tendencies and their relative contributions (Table S1 and Figure S3 in the Supplement) show that NMIX is even slightly higher than NCIR over Sahara and also over Europe, i.e. over highly polluted land. Thus, in these two regions the hierarchy found at the global scale changes to FREE > DETR > NMIX > NCIR > SECP.

Over the Amazon, significant convective activity boosts the importance of DETR relative to other regions. In both their vertical profiles and relative contributions, the two oceanic areas look similar despite being subject to different aerosol conditions. ICNC in these regions is also less frequently larger than $ICNC_{max}$, as the relative contribution of MINMAX- in IND_oce and in ATL_oce remains low (Figure S3). Finally, we note that the medians in Figure 8 and the statistics in Table S1 again indicate that the microphysical tendencies are characterised by skewed distributions. This is valid especially for FREE, whose median is close to zero (and not visible) in Figure 8.

## 5.5 Sensitivity studies

### 5.5.1 Impact of ice nucleation scheme

Having defined the hierarchy of the ICNC tendencies in REF, we continue now to analyse how microphysical parameterizations may change this hierarchy. We replace the ice nucleation parameterizations BN09 and P13 with the KL02 and LD06 schemes in the PRES simulation. A comparison of the tendencies between REF and PRES is given in Figure S4. As expected, the ice nucleation tendencies (i.e. NCIR and NMIX) exhibit the strongest differences; both increase in PRES, particularly NCIR, whose global mean increases by almost two orders of magnitude (Table 4). This jump in NCIR is due to the fact that KL02 parameterizes only homogeneous nucleation and disregards the competition for water vapour between homogeneous and heterogeneous nucleation and the effects of pre-existing ice crystals, producing more, smaller ICs than BN09. (A detailed comparison between the different ice nucleation parameterizations is given in Bacer et al. (2018).) The relative contribution of FREE also decreases, as the NCIR contribution increases (Figure 3), however, FREE remains the main IC source in terms of absolute values. The other source terms (DETR and SECP) do not change significantly: they decrease by less than 1% (Figure S4), and their global means are close to those computed in REF (Table 4). Overall, the application of different parameterizations for ice nucleation has only slightly changed the hierarchy of the IC sources (FREE > NCIR > DETR > NMIX > SECP in PRES).

Turning to the sinks, SELF, ACCR, and AGGR increase more than 5% in the upper troposphere (Figure S4), but their increase is still much smaller than that of the NCIR and NMIX sources. If many small ice crystals are produced, these sinks become much less efficient. The global means of the physical removal processes are almost unchanged in PRES with respect to REF (Table 4); however, we observe that the negative numerical tendencies strengthen and that the relative contribution of MINMAX- increases at the expense of ACCR and AGGR (Figure 3).

In conclusion, changing a given process parameterization can strongly influence that process tendency but may propagate weakly to other process tendencies. In particular, changing a source parameterization is expected to have only a small influence on the sink hierarchy. It is also important to note that, since parameterizations depend on model-computed quantities like vertical velocity and aerosol number concentrations as well as parameters like freezing threshold, tendencies are also strongly dependent on model setup.

### 5.5.2 Effects due to global warming

In order to estimate the global warming effect on cold cloud microphysical processes, we next compare the REF and FUT simulations. The relative percentage changes of the annual zonal means of the FUT tendencies with respect to the REF tendencies are displayed in Figure 9. Both microphysical tendencies for production and removal of ICs shift upward under global warming. As the surface temperature warms, the troposphere deepens and the lapse rate becomes less steep. Given the cold temperature criteria for most ICNC processes, their contributions must shift upward in altitude to reach the same temperature regime.

The DETR, SECP, AGGR, ACCR, and SELF tendencies all increase in magnitude (up to 10%) in the upper troposphere, while they slightly decrease (about 1%) at lower altitudes with warming. This is consistent with the upward shift of the

freezing levels indicated by the isotherms computed for FUT and in agreement with Del Genio et al. (2007). DETR increases at the highest levels in the tropics as overshooting convection may occur more often or extend deeper, carrying more ICs to these altitudes. In contrast, right around the homogeneous nucleation freezing level, DETR decreases. Upper-tropospheric static stability is expected to increase in a warmer climate, reducing the mass convergence into clear-sky regions and hence the ice detrainment (Bony et al., 2016). Indeed, we see a decrease in the mean upper-level divergence from the REF to the FUT simulation (Figure 10c), as well as a decrease in mean cloud fraction between 250 and 400 hPa across latitudinal bands (Figure 10b). While the detrainment increase above 200 hPa is driven by a few instances of extreme deep convection, the detrainment decrease around the melting layer is driven by mean convective behavior.

NCIR decreases in the upper troposphere. This can also be understood in terms of an increasing upper-tropospheric static stability, which dampens the vertical velocity and its subgrid component input to the ice nucleation scheme, both in the mean and at the 99th percentile (Figure 10d-e). With weaker vertical motion, less supersaturation is generated to drive ice nucleation. In spite of this decreased nucleation, we see an increase in overall ICNC between 200 and 300 hPa in the FUT simulation with respect to REF, both in absolute and relative differences (Figures S1 and 10a). This increase in upper-level ICNC manifests itself as an increase between 0.1 and 0.3 K day$^{-1}$ of the cloud longwave radiative heating in the FUT simulation (Figure 10f). This increased upper-level heating is important as it stabilizes the atmospheric column and suppresses deep convective activity.

Although we have not shown ice crystal radii here, if ICNC were to increase at a fixed cloud ice water content, the ICs would become smaller and their fall speeds would decrease. Decreased fall speed would, in turn, translate to more persistent ice clouds that warm the upper atmosphere over longer times. Also, while ICNC increases in a narrow vertical range between the homogeneous nucleation freezing level and tropopause, the global mean ICNC decreases by almost 30% in FUT relative to REF (Table 4); intuitively, a warmer future means less new ICs being produced and removed. At the global scale, the hierarchy of ICNC tendencies remains the same between the REF and FUT simulations.

## 6 Conclusions

We studied the relative importance of cold cloud microphysical process rates (tendencies) and the unphysical corrections (numerical tendencies) that affect ICNC using global simulations performed with the chemistry-climate model EMAC. The formation processes of ice crystals considered are ice nucleation in the cirrus regime (NCIR), ice nucleation in the mixed-phase regime (NMIX), secondary ice production (represented via the Hallet-Mossop process, SECP), convective detrainment (DETR), and instantaneous freezing of supercooled water cloud droplets (FREE). The loss processes of ice crystals are melting (MELT), self-collection (SELF), aggregation (AGGR), and accretion (ACCR); sublimation is excluded from the analysis of this study. We also evaluated the model in-cloud ICNC with satellite ICNC retrievals by the DARDAR-Nice data set. The comparison showed that EMAC reproduces the main features of the global ICNC distribution and the zonal ICNC profile, although there are differences in terms of absolute values. Like other models, EMAC overestimates ICNC in the cirrus regime in the extratropics, perhaps because of the instantaneous freezing process; on the other hand, ICNC is underestimated in the mixed-phase regime. One possible reason could be the low freezing threshold assumed for convective detrainment.

We analysed the global distributions and means of all microphysical tendencies, in particular defining a hierarchy of ice crystal sources and sinks. We found that, on average, the hierarchy of the IC sources is FREE > DETR > NCIR > NMIX > SECP, while the hierarchy of the IC sinks is AGGR > ACCR > SELF > MELT. The fact that freezing is the largest source of ICs, followed by detrainment, is in agreement with the results of Muench and Lohmann (2020), although they parameterized FREE differently, taking into account its dependence on updraft velocity (Kärcher and Seifert, 2016). Wernli et al. (2016) and Krämer et al. (2016) also found a predominance of liquid-origin cirrus over in-situ cirrus. We therefore reiterate that more efforts should be devoted to improve liquid-origin cirrus clouds (Muench and Lohmann, 2020). In the case of the CLOUD submodel, FREE consists of a direct conversion of cloud droplets into ICs, while it should not depend only on CDNC but also on updrafts, for example; thus, it is likely that FREE is overestimated in CLOUD. A better FREE parameterization should reduce the overestimation of ICNC with respect to observations, as indicated by our test simulation NOfree. The distributions of the tendencies are left-skewed. We found that the distribution of MELT is close to a bell-shaped distribution and the ones of SELF, AGGR, and ACCR are trimodal.

Numerical tendencies can have a non-negligible contribution to ICNC (Table 2 and Figure 3). The largest numerical tendency is negative and imposes an upper threshold of ICNC ($10^7 \mathrm{~m}^{-3}$). Our test simulation NOicncmax proved the strong effect of such numerical tendency in reducing ICNC. Working to reduce numerical tendencies is important because they could obscure the ice microphysical parameterization results. Such improvements would require using observations to infer active ice microphysical processes from, for example, crystal size distributions and the surrounding thermodynamic conditions and ensuring that the same processes are triggered in the model.

Regionally, the relative importance of the microphysical sources can vary, while the sinks appear similar. For example, heterogeneous nucleation in the mixed-phase regime is slightly more important than NCIR over the Sahara and Europe because of the abundance of INPs, while secondary ice production is more important than NMIX over the Amazon. Over the oceans, tendencies are similar even in different hemispheres, subject to different aerosol conditions.

Additionally, we found that different parameterizations for ice nucleation changed the ice nucleation tendencies but propagated only weakly to the other source and sink tendencies. Our sensitivity test suggests that the tendency hierarchy could change using different parameterizations for other microphysical processes but also another model setup. The large variation in ICNC output from the ice nucleation parameterizations corroborates the importance of including the competition for water vapor between INPs and pre-existing ice crystals (Bacer et al., 2018).

We also computed the tendencies in a future climate (using the RCP6.0 scenario). Our results shows an upward shift of the freezing level and the associated microphysical processes to higher altitudes, consistent with a reduced lapse rate and deepened troposphere that accompany surface temperature warming. Detrainment increases at the highest levels in the tropics, as overshooting convection may occur more often or extend deeper, in agreement with a decrease in the mean upper-level divergence, while it decreases around the homogeneous nucleation freezing level, where we found a decrease in mean cloud fraction across latitudinal bands. Ice nucleation decreases in the upper troposphere, due to weaker updrafts. Finally, we found an increase in upper-level ICNC in the FUT simulation causing an increase of the longwave radiative heating, which stabilizes the atmosphere. Globally, mean ICNC decreased by almost 30% in the warming scenario.

Knowing the relative importance of the microphysical process rates is of fundamental importance to assign priority to the development of microphysics parameterizations. Model improvements could benefit from the development of techniques that infer active ice microphysical processes from in-situ and remote sensing observations. Numerical tendencies can play a non-negligible role, and effort should be spent on minimizing the contribution of these. Moreover, the quantification of tendencies is essential to compare model output and observations which have different temporal resolutions.

In future studies about the relative importance of the cold cloud microphysical processes, it would be useful to perform a similar analysis for the mass tendencies, i.e. the rates of cloud ice mixing ratios. The tendencies of transport, sedimentation, and sublimation could also be included to close the ICNC budget. Finally, since cloud lifetime can be short, of the order of hours, it would be interesting to perform ensemble runs in order to test the sensitivity of the results to different output frequencies and also to various model resolutions.

*Data availability.*  The simulation data used in this study are available upon request.

*Author contributions.*  SB performed the model simulations with AP. SB analysed the data together with SS, OS, and AP. SB prepared the manuscript together with SS and SO. All the authors provided assistance in finalizing the analysis and the manuscript.

*Competing interests.*  The authors declare that they have no conflict of interest.

*Acknowledgements.*  The authors would like to thank the reviewers for their comments, which helped to improve the analysis in this study. Also, the authors would like to thank Steffen Münch from the ETH Zürich, Sergey Gromov and Klaus Klingmüller from the Max Planck Institute for Chemistry (MPIC), and Mattia Righi from the Deutsches Zentrum für Luft- und Raumfahrt (DLR) for the useful discussions. SS was funded by the DFG project TropiC in collaboration with NSF PIRE project 1743753. HT acknowledges funding from the Carl-Zeiss foundation. We acknowledge the usage of the Max Planck Computing and Data Facility (MPCDF) for the simulations performed in this work.

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

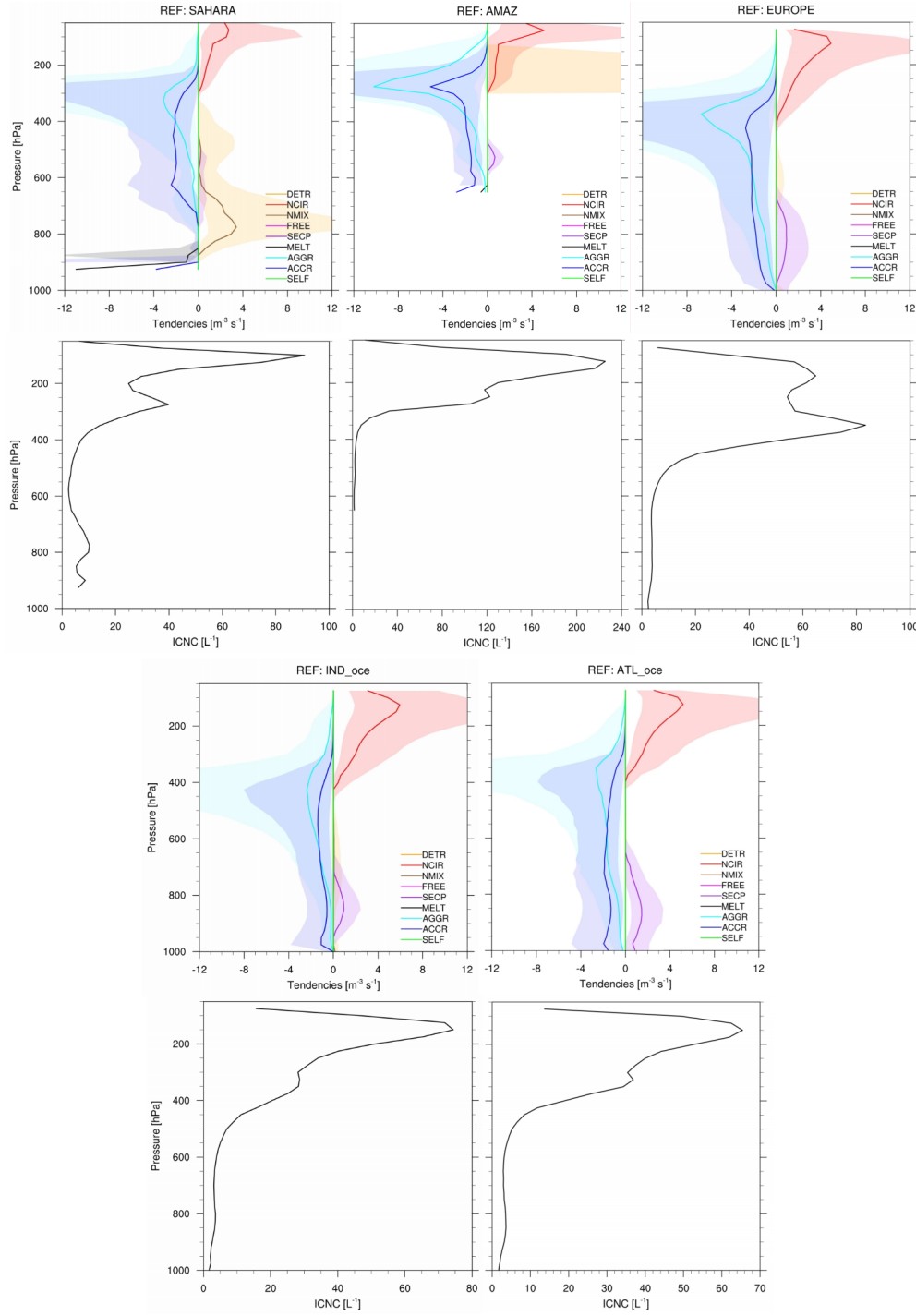

**Figure 8.** Microphysical process tendencies and ICNC as a function of pressure computed for different regions: Amazon, Sahara, Central Europe, Southern Indian Ocean, and North Atlantic Ocean. The vertical profiles are medians computed only where ICNC $> 1 \ \mathrm{L}^{-1}$ in bins of 25 hPa. The coloured shadows mark the areas between the 25th and the 75th percentiles.

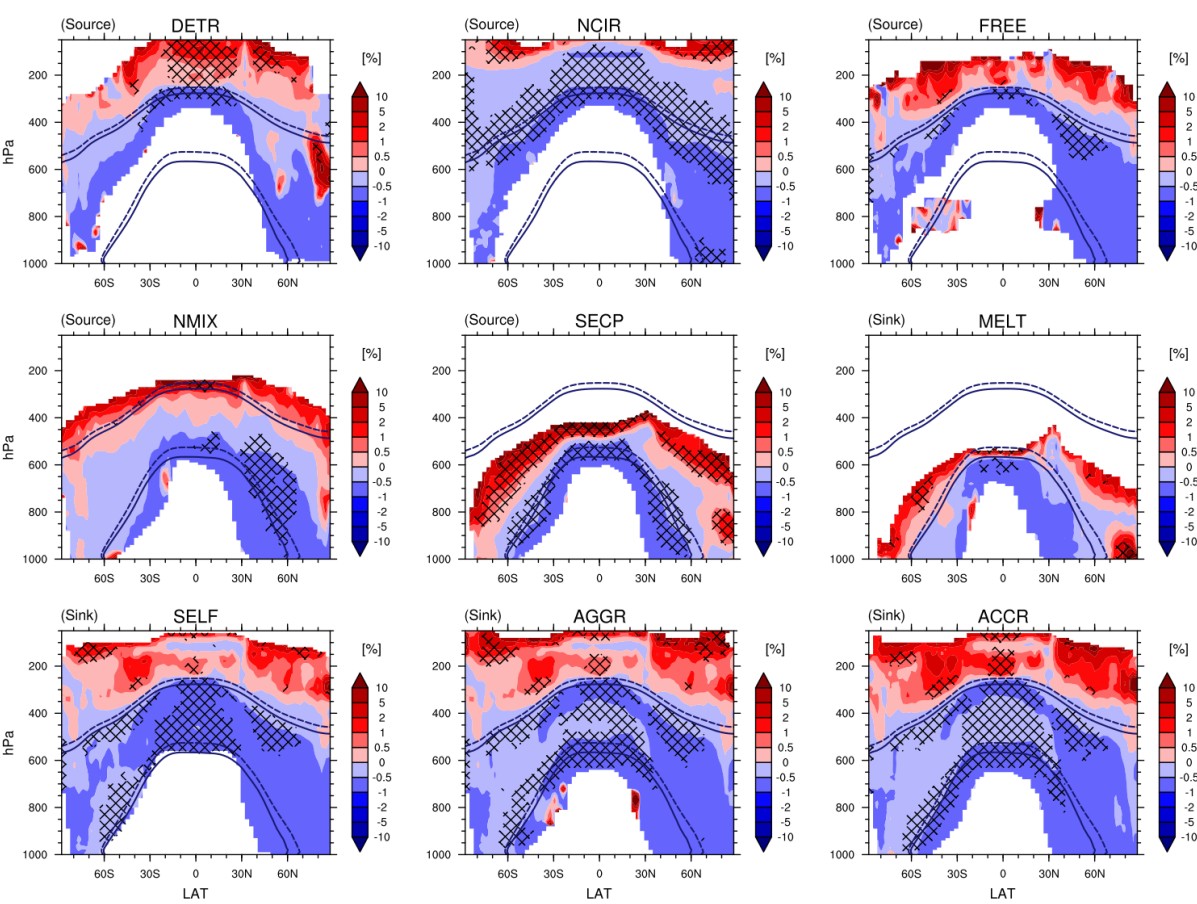

**Figure 9.** Percentage changes of the tendencies associated to ICNC microphysical processes in cold clouds in FUT with respect to REF. They are computed with daily means and are shown where REF daily means are $> 10^{-5} \mathrm{m}^{-3}\mathrm{s}^{-1}$. The hatched pattern indicates areas with a significance level of 90%. The isotherms at $0°\mathrm{C}$ and $-35°\mathrm{C}$ are annual means in REF (solid line) and in FUT (dashed line).

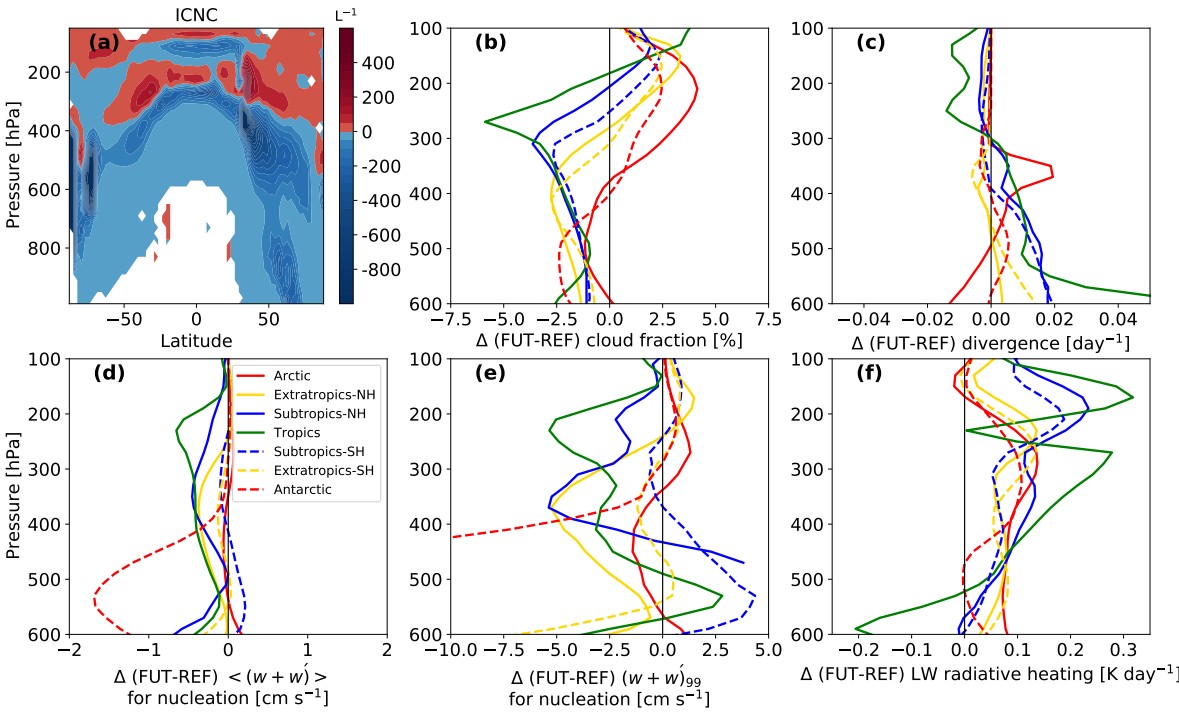

**Figure 10.** (a) Absolute difference in ICNC between the FUT and REF simulations. (b) FUT-REF differences in the mean cloud fraction; (c) mean divergence; (d) mean input vertical velocity to the ice nucleation scheme, grid-scale plus subgrid-scale variability term ($<w + w'>$); (e) extreme input vertical velocity to the ice nucleation scheme; and (f) longwave (LW) cloud radiative heating in different zonal bands. The Arctic is defined as north of $60°$N and the Antarctic south of $60°$S, the extratropics are between $40°$ and $60°$ S/N, the subtropics between $20°$ and $40°$ S/N, and the tropics between $20°$S and $20°$N.