# Peer review of "Figure S1. Annual means of vertically integrated ICNC (in $10^8 \text{ m}^{-2}$ ) for the simulation REF. The pink rectangles delimit the areas considered for the regional analyses in Section 4.3; the coordinates of the areas are specified in the table."

_Atmospheric Chemistry and Physics, 2020_

## Referee Comment (RC1) · Blaž Gasparini (Referee) · 31 May 2020

**REVIEW OF BACER ET AL., 2020, ACPD**

**Summary:**

The study tries to better understand cold clouds simulated by a GCM by showing the ice number microphysical rates both at global and regional scales. Surprisingly, freezing of cloud droplets was found to be the largest source of ice crystals, while snow formation by aggregation and accretion are the dominant sinks. The relative importance of sources and sinks does not change significantly with a different freezing scheme. Changes due to global warming are also briefly studied with the help of a separate model simulation.

The study is an interesting read and is novel given that it shows microphysical number rates, which are rarely shown or discussed in published literature and therefore represents a welcome expansion of the standard analysis of ice cloud properties. The study lacks some in-depth analysis on the reasons for the importance of some of the process rates and has some methodological issues. Given the unique perspective on cold clouds, I would encourage the authors to add a discussion on the potential weaknesses of the presented results and ideas for overcoming them in future model developing work on cold cloud processes.

Despite (or because of) its novelty, the presented analysis still has a large potential for improvement. I added a list of (1) general and (2) specific sources of concern and study weaknesses that the authors still need to address before the manuscript can be published.

**General comments:**

I. **Mass microphysical rates**
Why did you include only the number and not the mass rates in your analysis? The title suggests you are studying both mass and number rates. By including mass rates would be easier to get a more complete picture of how your model works. I assume the mass rate hierarchy could look quite different from the number rate hierarchy.

II. **Sublimation**
Why is sublimation not considered as a sink of ice number in the analysis, particularly after being mentioned in Eq. 1? I think it would be good to find a way to include sublimation in the analysis or at least estimate its impact.

III. **Closing the number budget and "numerical tendencies"**
How close are you to closing the number budget?
Are (sources+sinks)*model timestep = ICNC?

Your manuscript offers an often neglected insight into sources of ice, which is rarely seen in publications. However, you do not include "numerical tendencies" in the analysis. I think it would be valuable to show all numerical/unphysical tendencies (correction terms) that significantly perturb the ICNC budget besides the mentioned physical tendencies.
An example of such unphysical sink of ice (that you did not mention in the

manuscript) is the maximum ICNC correction term.

The ice cloud community should become more aware of all such terms and think about ways to avoid imposing such unphysical limits in the models of microphysics. By doing so, the ICNC picture would be complete, and you could close the sources and sink budget. This is in my opinion more important than limiting your analysis to the tendencies with physical meaning only. A strong additional message coming out of your work could therefore be that the very "volatile" ICNC budget is significantly modified by "numerical tendencies".

IV.  **FREE term**
I don't think you can physically justify the existence of FREE by simply referring to it as "liquid origin cirrus". The work of Krämer et al., 2016 associates liquid origin cirrus to deep convection (which is DETR in your case) or frontal ascent (e.g. warm conveyor belts). Wernli et al., 2016 shows a peak in liquid origin over the storm track region due to slow frontal ascent.

However, in your simulations, FREE is strikingly high over continents and orography. We know wave clouds could be formed by homogeneous freezing of cloud droplets (Heymsfield and Miloshevich, 1993), but that should not matter much in a climatic sense.

Homogeneous freezing of cloud droplets is to my understanding of ice cloud formation mechanisms climatically irrelevant outside of deep convective updrafts (and those are taken care of by deep convective scheme and DETR tendency).

I would therefore argue that one of the partly unphysical tendencies mentioned in the upper comment is your FREE term. I believe FREE is to a large extent just a temperature correction term that freezes the cloud droplets at temperatures <-35°C. Ideally, other processes in the model should take care of that and freeze most of the cloud droplets at warmer temperatures. Such terms appear also in other models. Do you believe we should be worried if they represent such a dominant source of ice? Why?

How would ICNC look like if you neglected the FREE tendency? Would a short experiment without the FREE source term help understanding its real climatic importance?
FREE, as you mention, does not happen very often, but results in huge ICNC. Therefore I would also expect the FREE term to often exceed the maximum ICNC threshold of $10^7$ m$^{-3}$ and therefore be immediately limited by the "maximum ICNC correction" IC sink. The net climatic effect of such a tendency may therefore be limited.

In summary of my lengthy comment, I believe the manuscript would benefit substantially if you better explored the causes of FREE.

V. **Relative importance of specific sources and sinks of ice**
It is hard to understand the relative importance of specific source and sink processes only by looking at the zonally averaged Fig. 2 and 3. Could you add plots showing the relative importance of each process, i.e. a division of a specific source or sink process with the total source or sink tendency.
Would a similar type of plot help in exploring the regional importance of several sources and sinks of ice in the discussion of Fig. 1 and Fig 4?

It may be easier to understand the importance of the separate microphysical rates if you would include also figures/information about:
   a. Probability density function distributions for each microphysical rate, plotted only when the rate has a non-zero value.
   b. Occurrence frequency of each of the microphysical rates.

VI. **SEDI tendency**
Why is the vertically integral of SEDI so negative? Shouldn't we think of sedimentation only as a redistribution of ice crystals? Shouldn't the column integrated net SEDI be equal to zero?

I know this is not possible due to the inclusion of sublimation of falling ice crystal into the sedimentation tendency. Could you therefore (1) analyse that tendency separately and (2) verify if the net SEDI is now close to be balanced.

I don't understand the reasoning you give explaining the disagreement between SEDI+ and SEDI- in lines 245-247. Isn't SEDI- in level X same as SEDI+ in level X-1? (if we take care for the sublimation of falling ice)

Moreover the median vertical profiles in Fig. 4 suggest that the vertical integral of sedimentation should be a small values, and not a significantly negative tendency as shown in Fig. 1. The zonally averaged perspective shows SEDI- being more dominant than SEDI+ at all levels of the atmosphere. Why is there such a disagreement between Fig 4 and Figs. 2+3?

VII. **Summary chart**
You could add a summary chart (maybe a pie chart for sinks and sources of ice or a bar chart) that summarizes the importance of several sources and sink processes. Table 3 is to some extent doing that, but tables are hard to read (and also table 3 is not really giving us a budget perspective). I think that such a visualization (maybe in relative, not absolute terms) would be a nice key figure of the paper.

VIII. **Effects due to global warming**
Section 4.4.2 is currently very weak and doesn't really provide much of robust novel findings. The only robust feature is the upward shift of ice rates/ICNC. The changes to ICNC, IWC, IWP, source and sink processes cannot be considered robust when comparing only 1 year of data (!). This is confirmed by no significance in zonally averaged plots (I don't consider a 70% significance level adequate).

The upward shift in clouds (and therefore sources/sinks of ice) is not novel, so

I suggest removing the section and rather focus on digging more into the model to better understand the above mentioned points.

If you really want to keep it, you should substantially expand your analysis. A climate change or cloud feedback perspective on the shifts of ice phase with global warming would certainly need some new plots, e.g. changes in ICNC, IWC, IWP, specific and relative humidity, a cloud feedback decomposition (or at least changes in cloud radiative effects assuming an adjustment term to take into account changes in clear sky quantities/changes between a CRE and a cloud feedback perspective). Maybe also changes in static stability, radiative heating, etc.

Moreover, you did not take advantage of the high frequency output data. How does the ICNC distribution shifts (a) in total (b) in specific temperature ranges? What about IC sources and sinks?

**Specific comments:**

➢ Line 4:
  How could you compare microphysical process rates with observations? Sadly, I think it's hard to measure the relevant number process rates with the available in-situ or remote sensing data. Observations currently lack the evolution perspective, and rather give snapshots of cloud properties.

➢ Line 13:
  You could verify whether cloud diabatic heating rates increase in the upper troposphere with the additional model diagnostics.

➢ Introduction: A reference mentioning the work by Dietlicher et al., 2019 who showed the cloud volume based on source may be appropriate, although the distinction is not necessarily a process-rate based one. A reference to Gyrspeerd et al., 2018 may also be appropriate given their cirrus classification scheme.

➢ I would find it useful if you started the paper by showing the ICNC zonal average and ICNC burden plots (S1 and S2a) and compare that with observations (Sourdeval et al., 2018 and Gryspeerdt et al., 2018). Why does your model overestimate ICNC in the extratropics while simulating too little ICNC in the tropics?

➢ Section 2.1/2.2:
  Is snow diagnostic? Is it removed from the atmosphere in one timestep? Does it affect radiation or not?

➢ Line 123:
  The convective scheme should detrain some ice also at temperatures warmer than -35°C. A recent publication by Coopman et al., 2020, for example, shows that the average glaciation temperature of isolated convective clouds over Europe is about -21°C. That may be worth mentioning in the text as a potential problem of the

scheme and reason for low ICNC bias in mixed phase compared to observational data by Sourdeval et al., 2018.

➢ Section 2.3:
Please describe how each of the IC sinks works (not only refer to older publications, given the central role of such processes in your paper).
Is there a temperature dependence (particularly for aggregation, accretion, and self collection)?
Is there any size dependence?

➢ Section 3:
Do you run your global warming simulation in present-day $CO_2$ concentrations?
If so - do you expect any influence from not changing $CO_2$ levels to those expected in year 2080 in the RCP6.0 scenario?

➢ Lines 230-233:
*In fact, upper-level gravity wave activity, particularly strong in the tropics, can generate temperature fluctuations responsible for strong nucleation tendencies.*

Is this right? Your model resolution is about 3°x 3°, which is orders of magnitude larger than the relevant length scales for gravity waves. So the model cannot resolve those directly.
Moreover, the model used doesn't seem to have a parameterization that would add a gravity wave updraft spectrum in to the vertical velocity and in such way represent the influence of gravity waves on ice nucleation. I guess the used TKE-based updraft only gives one vertical velocity value per gridbox, not a distribution.
I believe the reason for high ice nucleation rates in the tropical upper troposphere therefore lies in a combination of cold temperature and high relative humidity.

➢ Line 240:
*On the contrary, SEDI+ is low at upper levels because the crystals are too small to fall out and at lower levels because the number of ICs is a small.*

That doesn't sound right or I simply don't understand it. Wouldn't that be true for SEDI- and not SEDI+.
ICs are small at upper levels, but I don't know why this would limit the SEDI+ tendency. I would assume SEDI+ tendency to be larger in locations where ICNC is large and where IC radius is small. This points rather at the upper troposphere.

➢ Section 4.3 and Fig 4.

Why is detrainment so important over Sahara? Why at such higher altitude?
I assume the number of points taken for the Sahara figure is small due to the low amount of ice clouds there. That may be added in the discussion.

Another general conclusion of this section could be that a clean (southern) Indian Ocean is very similar to a more polluted N. Atlantic?
Moreover, I think many atmospheric scientists would rather call that region as

"Southern Ocean", as in this large project
https://www.eol.ucar.edu/field_projects/socrates , for example. When talking about Indian Ocean we normally think of tropics.

I still cannot understand whether FREE is an important source of ice or not. A sensitivity experiment in which FREE source would be turned off could help determining that by looking at changes to ICNC.

➤ Line 310: Is DETR really maximal at -35°C? I cannot see that from the Fig. 2. Detrainment tendency should be probably maximized at temperatures closer to -50°C (220 K) in the tropics, if we believe the FAT theory (Hartmann and Larson, 2002).

➤ Section 4.4.2:
It is hard to understand whether we see only a shift or some change in ice rates. A temperature vertical axis would therefore be more appropriate for Fig 5.

➤ Line 313-320:
You mention the ICNC increase in the upper troposphere. Isn't this only a shift due to the expansion of troposphere?
If IC radius decreases and if this change is important, you may want to show it in a separate plot.

➤ Line 318:
Isn't an increase in cloud persistence in contrast with your comment on decreased upper tropospheric anvil clouds due to increased static stability? I thought the high cloud fraction decreases with warming?

➤ Line 315:
Why do you think the LW atmospheric heating is associated with the cloud base temperature? Are you talking about heating within the atmosphere? Or at the top-of-the atmosphere (TOA) radiative effects?
I don't think the cloud base temperature matters for the TOA LW effects. Deep convective clouds have a large LW CRE, despite having a very low cloud base (with high temperatures). Maybe some of Mark Zelinka's numerous publications on the topic may help.

➤ Line 315:
Also, you talk about the additional upper tropospheric warming due to climate change but never explain why should we care if the upper troposphere is slightly warmer? (compared to the arguably more important or at least more studied influence of changes in high clouds on the TOA radiative budget and climate sensitivity)

➤ Line 315-316:
I think the sentence "thicker cirrus…" is incorrect. Why only thick cirrus? Also, most cirrus aren't optically very thick.

➤ Lines 319-320:

I am not sure if the interpretation of the result of Sanderson et al., 2008 is correct, so it may need to be rewritten. Sanderson et al., 2008 found the IC fall speed to be important in modulating the mainly LW cloud feedback (and hence climate sensitivity) not because the IC fall speed would change between the present day and global warming simulation (IC fall speed is not calculated interactively in their simulations, given the use of a tuning parameter). However, a smaller ice fall speed leads to more high clouds. That in turn leads to a larger LW altitude (positive) cloud feedback, which is the dominant high cloud feedback. On the other hand, a smaller present-day cloud fraction due to large ice fall speed, leads to less high clouds and a smaller high cloud feedback and smaller climate sensitivity.

➢ As you talk about ice clouds and not only cirrus, you may want to also explore/mention the cloud phase negative optical feedback due to global warming (Tan et al., 2016, maybe also Bodas-Salcedo 2018 and 2019, Lohmann and Neubauer, 2018).

➢ Conclusion (in general):
It may be appropriate to think a bit more about some of the questions I listed below and include some of that in the discussion:

What did you learn about the model by exposing the number tendencies that you couldn't by simply taking the ICNC fields?
Is there something that we should be worried about? Why? What is causing it?
What are the potential weaknesses of the study? How does this compare to other work (if any exists – maybe for mass rates)?

**References**

- Bodas-Salcedo, 2018: Cloud Condensate and Radiative Feedbacks at Midlatitudes in an Aquaplanet
- Bodas-Salcedo, 2019: Strong Dependence of Atmospheric Feedbacks on Mixed‑Phase Microphysics and Aerosol‑Cloud Interactions in HadGEM3
- Dietlicher et al., 2019: Elucidating ice formation pathways in the aerosol-climate model ECHAM6-HAM2
- Gryspeerdt et al., 2018: Ice crystal number concentration estimates from lidar-radar satellite remote sensing. Part 2: Controls on the ice crystal number concentration
- Gryspeerdt et al., 2018: Technical note : An automated cirrus classification
- Hartmann and Larson, 2002: An important constrain on tropical cloud-climate feedback
- Lohmann and Neubauer 2018: The importance of mixed-phase and ice clouds for climate sensitivity in the global aerosol–climate model ECHAM6-HAM2
- Sanderson et al., 2008: Toward Constraining Climate Sensitivity by Linear Analysis of Feedback Patterns in Thousands of Perturbed-Physics GCM Simulations
- Sourdeval et al., 2018: Ice crystal number concentration estimates from lidar-radar satellite remote sensing - Part 1: Method and evaluation

- Tan et al., 2016: Observational constraints on mixed-phase clouds imply higher climate sensitivity

---

## Referee Comment (RC2) · Anonymous Referee #2 · 20 Jun 2020

This study quantifies and investigates the cold cloud microphysical process rates using one chemistry-climate model EMAC, and defines the hierarchy of sources and sinks of ice crystals. The analysis is carried out both at global and at regional scales. It is an interesting idea to quantify the important ice crystal sources and sinks globally so this is the work worthy of being published. However, before it can be accepted, there are a few major concerns to be addressed.

(1) The model result uncertainty could be very large from a few aspects.

1.1 The model grid spacing is very coarse ($\sim$300 km) and the output time frequency is very sparse (every 5 hours). Many times, the cloud lifetime can be even less than 10 hours, then the sampling cannot be representative with every 5-hour time frequency. I'd suggest look at the sensitivity to model resolution (such as 100 km) and output time

frequency (hourly) to meet the goal of quantification.

1.2 Need to do ensemble runs for quantification.

1.3 Need to discuss that the results might be changed with different models or different physical parameterizations such as cumulus or microphysics parameterizations.

(2) For the sink of ice crystal, sublimation should be considered.

(3) Result section: I feel a little surprised that the authors started the discussion of results for the source and sink of ice directly. It would be nice to understand the overall model performances in simulating radiation, clouds and precipitation first. Then get to the analysis of ice crystal number concentrations and its budget.

(4) Since one of the purposes of the study is to test the sensitivity to two other nucleation parameterizations, then some description about the two default and two tested schemes is needed, particularly about how different they are in terms of representing ice formation such as temperature dependent, supersaturation dependent, and aerosol dependent. If aerosol dependent, then what aerosols are considered? Why did you replace the immersion freezing scheme with a contact freezing scheme? Shouldn't they be considered together?

Minor comments,

1. Calling everything below -35 deg C as "cirrus clouds" is not accurate. I would suggest change to "pure ice clouds".

2. For the convective detrainment, does the model treat the detrainment at the levels with T> -35 deg C? If not, is there a reason? Theoretically convective detrainment of droplet and ice can occur from middle to top troposphere.

3. Line 210-215, does FREE include the droplet freezing in convective parameterization?

4. Section 4.2, how to reconcile that DETR is much larger than NCIR in zonal mean

(Fig. 2) but smaller than it in global spatial distribution (Fig. 1)?

5. Line 284-286, I am confused by this sentence. Earlier it is said LD06 is a contact freezing scheme which is for heterogenous freezing. Here you said LD06 parameterizes only homogeneous nucleation. Also P13 should be an immersion freezing scheme which should be much more efficient than the contact freezing LD06, but the results in section 4.4.1 did not even mention the differences they can make.

---

## Author Comment (AC1) · 27 Nov 2020

**Authors' reply to Referee #1**

We really thank Referee #1, Dr. Blaž Gasparini, for his constructive and helpful comments. Below, we provide our replies to his comments; number of lines and sections refer to the old manuscript.

**General Comments**

**I) Mass microphysical rates**

*Why did you include only the number and not the mass rates in your analysis? The title suggests you are studying both mass and number rates. By including mass rates would be easier to get a more complete picture of how your model works. I assume the mass rate hierarchy could look quite different from the number rate hierarchy.*

Since our previous study (Bacer et al. 2018), we have focused on the number concentration of ice crystals (ICs). This work follows up on the same direction, therefore, only the rates of ICNCs have been identified in the CLOUD submodel, saved, and analysed. Currently, the mass rates are not output variables, and it is not possible to include them in our analysis. We agree that the mass rate analysis would be interesting and, in fact, this was written at the end of the Conclusions.

We would like to keep this title (also Gettelman et al. 2013, who dealt with mass rates, used a general title "Microphysical process rates and global aerosol-cloud interactions"). Nevertheless, we agree that the reader should understand soon that the paper will focus on number rates, therefore, we made this clear in the Abstract.

**II) Sublimation**

*Why is sublimation not considered as a sink of ice number in the analysis, particularly after being mentioned in Eq. 1? I think it would be good to find a way to include sublimation in the analysis or at least estimate its impact.*

In the CLOUD submodel, sublimation is taken into account as an IC sink, but it is not dealt as an independent term. Inside the code, sublimation can only affect sedimentation, this is the reason why we wrote that "*SEDI includes also the sublimation of falling ICs*" at L172. The separation between sublimation and sedimentation is not straightforward and we cannot include it in this study.

**III) Closing the number budget and "numerical tendencies"**

*How close are you to closing the number budget? Are (sources+sinks)\*model timestep = ICNC?*

*Your manuscript offers an often neglected insight into sources of ice, which is rarely seen in publications. However, you do not include "numerical tendencies" in the analysis. I think it would be valuable to show all numerical/unphysical tendencies (correction terms) that significantly perturb the ICNC budget besides the mentioned physical tendencies. An example of such unphysical sink of ice (that you did not mention in the manuscript) is the maximum ICNC correction term. The ice cloud community should become more aware of all such terms and think about ways to avoid imposing such unphysical limits in the models of microphysics. By doing so, the ICNC picture would be complete, and you could close the sources*

[Figure]

Figure 1: Annual cycles of monthly means of vertically integrated ICNC global means. Continuous lines refer to the REF simulation; the dashed line refers to the test-simulation without the maximum ICNC threshold (NOicncmax).

*and sink budget. This is in my opinion more important than limiting your analysis to the tendencies with physical meaning only. A strong additional message coming out of your work could therefore be that the very "volatile" ICNC budget is significantly modified by "numerical tendencies".*

The number budget cannot be closed in this study because the advective, turbulent, and convective transport tendencies are not taken into account in our analysis (as written at L153). Moreover, in order to close the budget for in-cloud ICNC, the time integration using the Asselin filter should be applied.

What we checked was the validity of the following equality at a given timestep:

$$\sum R_i = (ICNC_{final} - ICNC_{initial})/\delta t$$

where $R_i$ are all ICNC tendencies detected in CLOUD (both physical and numerical), $ICNC_{initial}$ is the ICNC input value for CLOUD, $ICNC_{final}$ is the updated value of ICNC in CLOUD, $\delta t$ is the model time step.

In order to show, approximately, that the ICNC budget in EMAC is closed, we computed the annual cycles of the vertically integrated ICNC global means (Figure 1 in this document); it is evident that all years show the same behaviour, without any statistical trend.

In the old manuscript, we focused on the physical tendencies (as written at L173-176) and mentioned the existence of the numerical tendencies, providing the example of the maximum correction term, i.e. the threshold $10^7$ m$^{-3}$ for ICNC. Nevertheless, we agree with the Referee that including the analysis of the numerical tendencies would yield the awareness of the potentially important role of the numerical tendencies in computing ICNC. Therefore, we included this analysis in the revised manuscript.

Additionally, a test-simulation (NOicncmax) was run by removing the condition that ICNC must not exceed $10^7$ m$^{-3}$. Figure 1 shows the strong impact of this condition on ICNC, whose values are much higher than ICNC in REF.

Some new text regarding the numerical tendencies and the new test-simulation has been

added in the revised manuscript in the Abstract (L6), Introduction (L76-79), Section 2 (with the addition of a new subsubsection), Section 3, Section 5, and Conclusions.

*IV) FREE term*
*I don't think you can physically justify the existence of FREE by simply referring to it as "liquid origin cirrus". The work of Krämer et al., 2016 associates liquid origin cirrus to deep convection (which is DETR in your case) or frontal ascent (e.g. warm conveyor belts). Wernli et al., 2016 shows a peak in liquid origin over the storm track region due to slow frontal ascent. However, in your simulations, FREE is strikingly high over continents and orography. We know wave clouds could be formed by homogeneous freezing of cloud droplets (Heymsfield and Miloshevich, 1993), but that should not matter much in a climatic sense. Homogeneous freezing of cloud droplets is to my understanding of ice cloud formation mechanisms climatically irrelevant outside of deep convective updrafts (and those are taken care of by deep convective scheme and DETR tendency). I would therefore argue that one of the partly unphysical tendencies mentioned in the upper comment is your FREE term. I believe FREE is to a large extent just a temperature correction term that freezes the cloud droplets at temperatures $<-35°C$. Ideally, other processes in the model should take care of that and freeze most of the cloud droplets at warmer temperatures. Such terms appear also in other models. Do you believe we should be worried if they represent such a dominant source of ice? Why?*
*How would ICNC look like if you neglected the FREE tendency? Would a short experiment without the FREE source term help understanding its real climatic importance? FREE, as you mention, does not happen very often, but results in huge ICNC. Therefore I would also expect the FREE term to often exceed the maximum ICNC threshold of $10^7 \ m^{-3}$ and therefore be immediately limited by the "maximum ICNC correction" IC sink. The net climatic effect of such a tendency may therefore be limited. In summary of my lengthy comment, I believe the manuscript would benefit substantially if you better explored the causes of FREE.*

Although FREE is represented in the model simply (like a condition which converts into ICs those cloud droplets that are transported in regions where temperature is below the freezing threshold), its inclusion in cloud microphysics schemes goes back at least to Levkov et al. (1992), as far as we know. According to Krämer et al. 2016, liquid-origin cirrus are formed by water droplets that freeze spontaneously when they reach the homogeneous freezing threshold. This is also the definition of FREE in EMAC, and assigning the meaning of FREE to liquid-origin cirrus is in agreement with Wernli et al. 2016 and Muench and Lohmann (2020). Therefore, we think that it is correct to treat FREE as a microphysical tendency and not a numerical tendency. Moreover, Muench and Lohmann (2020) also considered the freezing of cloud droplets as a source of ICs, although they developed the representation of such process considering its dependence on updraft velocity. More precisely, they analysed the following sources of ICs: homogeneous nucleation (our NCIR), heterogeneous nucleation in cirrus clouds (which is included in our NCIR as well), heterogeneous nucleation in mixed-phase clouds (our NMIX), convective detrainment (our DETR), and droplet freezing (our FREE). The global distribution and the zonal mean of their freezing are similar to our results. We discussed these points in the revised manuscript (in Section 2.2, in the analysis of the results in Section 5, and in Conclusions).

In order to investigate in more depth the role of FREE (as suggested by the Referee),

we performed another test-simulation (NOfree) where the tendency FREE is neglected. The description of the new simulation NOfree and its analysis have been added in Section 3 and in the new Section 5.1, respectively. In summary, we found that the tendencies in NOfree remain similar to the ones computed in the REF simulation, but ICNC globally decreases by one order of magnitude and CDNC instead increase by 10%.

*V) Relative importance of specific sources and sinks of ice*
*It is hard to understand the relative importance of specific source and sink processes only by looking at the zonally averaged Fig. 2 and 3. Could you add plots showing the relative importance of each process, i.e. a division of a specific source or sink process with the total source or sink tendency. Would a similar type of plot help in exploring the regional importance of several sources and sinks of ice in the discussion of Fig. 1 and Fig 4?*
*It may be easier to understand the importance of the separate microphysical rates if you would include also figures/information about: a) Probability density function distributions for each microphysical rate, plotted only when the rate has a non-zero value. b) Occurrence frequency of each of the microphysical rates.*

We computed the occurrence of each microphysical process considering non-zero values (new Figure 3 in the manuscript). For an easier comparison between processes, we preferred not to normalize the counts (to get a PDF). Moreover, we computed the relative contributions of the mean tendencies, and we represented them in pie charts. The relative importance was computed for the global means (new Figure 2) and for the regional means (new Figure S3 in the Supplement). We would like to stress the new Table 4 contains also the means and the standard deviations for the two new test-simulations. Since the distributions of the tendencies are described in the new Figure 3, the 1th and 99th percentiles were removed from the Table. The new figures are commented in the new Subsection 5.1 "Global statistics".

*VI) SEDI tendency*
*Why is the vertically integral of SEDI so negative? Shouldn't we think of sedimentation only as a redistribution of ice crystals? Shouldn't the column integrated net SEDI be equal to zero? I know this is not possible due to the inclusion of sublimation of falling ice crystal into the sedimentation tendency. Could you therefore (1) analyse that tendency separately and (2) verify if the net SEDI is now close to be balanced. I don't understand the reasoning you give explaining the disagreement between SEDI+ and SEDI- in lines 245-247. Isn't SEDI- in level X same as SEDI+ in level X-1? (if we take care for the sublimation of falling ice)*
*Moreover the median vertical profiles in Fig. 4 suggest that the vertical integral of sedimentation should be a small values, and not a significantly negative tendency as shown in Fig. 1. The zonally averaged perspective shows SEDI- being more dominant than SEDI+ at all levels of the atmosphere. Why is there such a disagreement between Fig 4 and Figs. 2+3?*

SEDI is a vertical redistribution of ICs (as written at L170). The vertical integration of SEDI in Fig. 1 was not zero because it was (wrongly) computed with monthly means. While using monthly means for the other tendencies is correct because each tendency has only positive or negative sign (and the mean computed with monthly means is equal to the mean computed with original output data), it was a mistake to use monthly means to compute the vertical integration of SEDI. In order to get a vertical integration of SEDI close to zero we have to use instantaneous values, as in Figure 2 (in this document). Nevertheless, SEDI

[Figure]

Figure 2: Vertical sum of instantaneous values of SEDI at one model time step (in $10^5$ m$^{-2}$s$^{-1}$).

cannot be exactly zero because of the inclusion of IC sublimation (as written at point *II*), sublimation affects sedimentation) and because SEDI is a net sink close to the ground (at the lowermost model level).

Since sedimentation is not a microphysical process but is a redistribution of existing ICs, we decided to remove the analysis of SEDI in the revised manuscript.

Regarding question about the vertical profiles in Fig.4, it must be taken into account that they are (median values) computed only where ICNC $> 1$ L$^{-1}$, while the plots in Figs. 2+3 are means computed without any mask, thus, Fig.4 and Figs. 2+3 are not directly comparable. More precisely, in Fig.4, positive and negative values of SEDI cannot be balanced because the statistics is computed for a total number of points which changes at each vertical 20 hPa-bin due to the application of the mask (ICNC $> 1$ L$^{-1}$) at each bin.

*VII) Summary chart*
*You could add a summary chart (maybe a pie chart for sinks and sources of ice or a bar chart) that summarizes the importance of several sources and sink processes. Table 3 is to some extent doing that, but tables are hard to read (and also table 3 is not really giving us a budget perspective). I think that such a visualization (maybe in relative, not absolute terms) would be a nice key figure of the paper.*

Please, see our reply to point *V)*.

*VIII) Effects due to global warming*
*Section 4.4.2 is currently very weak and doesn't really provide much of robust novel findings. The only robust feature is the upward shift of ice rates/ICNC. The changes to ICNC, IWC, IWP, source and sink processes cannot be considered robust when comparing only 1 year of data (!). This is confirmed by no significance in zonally averaged plots (I don't consider a 70% significance level adequate).*
*The upward shift in clouds (and therefore sources/sinks of ice) is not novel, so I suggest removing the section and rather focus on digging more into the model to better understand the above mentioned points. If you really want to keep it, you should substantially expand*

*your analysis. A climate change or cloud feedback perspective on the shifts of ice phase with global warming would certainly need some new plots, e.g. changes in ICNC, IWC, IWP, specific and relative humidity, a cloud feedback decomposition (or at least changes in cloud radiative effects assuming an adjustment term to take into account changes in clear sky quantities/changes between a CRE and a cloud feedback perspective). Maybe also changes in static stability, radiative heating, etc. Moreover, you did not take advantage of the high frequency output data. How does the ICNC distribution shifts (a) in total (b) in specific temperature ranges? What about IC sources and sinks?*

We thank the Referee for his feedback. We have strengthened the section on global warming by comparing five years of data in the reference period to five years in the warming period and adding new analysis.

**Minor Comments**

**L4:** *How could you compare microphysical process rates with observations? Sadly, I think it's hard to measure the relevant number process rates with the available insitu or remote sensing data. Observations currently lack the evolution perspective, and rather give snapshots of cloud properties.*

We are limited in such an observational comparison, since it is not straightforward to infer the process by which an ice crystal was formed (shape, size, proximity to convection and aerosol source). This additional data is not available at the global scale.

**L13:** *You could verify whether cloud diabatic heating rates increase in the upper troposphere with the additional model diagnostics.*

We verified this with the new Figure 10, where we do indeed see that the longwave radiative heating associated with ice clouds increases by 0.2-0.3 K per day in the upper atmosphere.

**Intro:** *A reference mentioning the work by Dietlicher et al., 2019 who showed the cloud volume based on source may be appropriate, although the distinction is not necessarily a process-rate based one. A reference to Gyrspeerd et al., 2018 may also be appropriate given their cirrus classification scheme.*

We added the first reference in the Introduction and the second one in the new Section 4 (see next point).

**Sec. 4:** *I would find it useful if you started the paper by showing the ICNC zonal average and ICNC burden plots (S1 and S2a) and compare that with observations (Sourdeval et al., 2018 and Gryspeerdt et al., 2018). Why does your model overestimate ICNC in the extratropics while simulating too little ICNC in the tropics?*

We included Figures S1, S2a and two new plots for in-cloud ICNC retrieved from satellite products (we used the DARDAR data set) in the new Section 4 "*ICNC model results and evaluation*". We discussed in the Conclusions that FREE could cause an overestimation of ICNC.

**Secs. 2.1-2.2:** *Is snow diagnostic? Is it removed from the atmosphere in one timestep? Does it affect radiation or not?*

Snow (precipitation) is fully diagnostic. Vertical advection of snow is not explicitly calculated, and snow reaches the ground in the same time step in which it is formed. Snow, which is not a prognostic variable nor a 3D variable, does not interact with the radiation scheme.

L123: *The convective scheme should detrain some ice also at temperatures warmer than -35 °C. A recent publication by Coopman et al., 2020, for example, shows that the average glaciation temperature of isolated convective clouds over Europe is about -21 °C. That may be worth mentioning in the text as a potential problem of the scheme and reason for low ICNC bias in mixed phase compared to observational data by Sourdeval et al., 2018.*

We thank the referee for the interesting reference; we cited it in Section 4.2.

Sec. 2.3: *Please describe how each of the IC sinks works (not only refer to older publications, given the central role of such processes in your paper). Is there a temperature dependence (particularly for aggregation, accretion, and self collection)? Is there any size dependence?*

We expanded the paragraph "Sinks of ice crystals", replying to the Referee's questions.

Sec. 3: *Do you run your global warming simulation in present-day CO2 concentrations? If so, do you expect any influence from not changing CO2 levels to those expected in year 2080 in the RCP6.0 scenario?*

The CO2 emissions in the FUT simulation are taken from the RCP6.0 scenario (see L198 in the manuscript), therefore, the results of FUT already include the influence due to CO2 level changes.

L230-233: *"In fact, upper-level gravity wave activity, particularly strong in the tropics, can generate temperature fluctuations responsible for strong nucleation tendencies."*
*Is this right? Your model resolution is about 3°x 3°, which is orders of magnitude larger than the relevant length scales for gravity waves. So the model cannot resolve those directly. Moreover, the model used doesn't seem to have a parameterization that would add a gravity wave updraft spectrum in to the vertical velocity and in such way represent the influence of gravity waves on ice nucleation. I guess the used TKE-based updraft only gives one vertical velocity value per gridbox, not a distribution. I believe the reason for high ice nucleation rates in the tropical upper troposphere therefore lies in a combination of cold temperature and high relative humidity.*

We removed this sentence and we changed L229-232.

L240: *"On the contrary, SEDI+ is low at upper levels because the crystals are too small to fall out and at lower levels because the number of ICs is a small."*
*That doesn't sound right or I simply don't understand it. Wouldn't that be true for SEDI- and not SEDI+. ICs are small at upper levels, but I don't know why this would limit the SEDI+ tendency. I would assume SEDI+ tendency to be larger in locations where ICNC is large and where IC radius is small. This points rather at the upper troposphere.*

We removed this sentence as we do not consider sedimentation in the revised manuscript.

Sec. 4.3 Fig.4: *Why is detrainment so important over Sahara? Why at such higher altitude? I assume the number of points taken for the Sahara figure is small due to the low amount of ice clouds there. That may be added in the discussion.*

*Another general conclusion of this section could be that a clean (southern) Indian Ocean is very similar to a more polluted N. Atlantic? Moreover, I think many atmospheric scientists would rather call that region as "Southern Ocean", as in this large project https://www.eol.ucar.edu/field_projects/socrates , for example. When talking about Indian Ocean we normally think of tropics.*

*I still cannot understand whether FREE is an important source of ice or not. A sensitivity experiment in which FREE source would be turned off could help determining that by looking at changes to ICNC.*

In the new profiles over Sahara, which are obtained with 5-years data (instead of 1-year data) of the new simulations considering bins of 25 hPa (instead of 20 hPa), the DETR profile is not visible anymore. According to Table S1 and the new Figure S3, DETR is more important over Amazon than over Sahara. We added some comments, also regarding the profiles over ocean, in Section 4.3.

The region identified as "IND_oce" is between the Southern Indian Ocean and the Southern Ocean; we specified in the text that with "IND_oce" we actually mean a region in the Southern Indian Ocean.

We performed a test-simulation without FREE; please, see our reply to point *IV)*.

L310: *Is DETR really maximal at -35°C? I cannot see that from the Fig. 2. Detrainment tendency should be probably maximized at temperatures closer to -50°C (220 K) in the tropics, if we believe the FAT theory (Hartmann and Larson, 2002).*

We changed this line in the revised manuscript.

Sec. 4.4.2: *It is hard to understand whether we see only a shift or some change in ice rates. A temperature vertical axis would therefore be more appropriate for Fig 5.*

For consistency with the other zonal mean plots, we have chosen to keep the pressure level coordinate for this figure.

L313-320: *You mention the ICNC increase in the upper troposphere. Isn't this only a shift due to the expansion of troposphere? If IC radius decreases and if this change is important, you may want to show it in a separate plot.*

We see both an upward shift with the changing atmospheric temperature structure (deepening troposphere as said by the Referee) as well as an increase in the ICNC tendency magnitudes. The colorbar in Figure 5 is symmetric and we see larger magnitude increases than decreases. The IC radius is unfortunately not a default output of these simulations.

L318: *Isn't an increase in cloud persistence in contrast with your comment on decreased upper tropospheric anvil clouds due to increased static stability? I thought the high cloud fraction decreases with warming?*

Yes, we thank the Referee for pointing this out. We have clarified that such a mechanism would counterbalance those associated with increased static stability. Since we see large

decreases in upper-level cloud fraction, our results do not support such a "decreased-fall-speed" mechanism.

L315: *Why do you think the LW atmospheric heating is associated with the cloud base temperature? Are you talking about heating within the atmosphere? Or at the top-of-the atmosphere (TOA) radiative effects? I don't think the cloud base temperature matters for the TOA LW effects. Deep convective clouds have a large LW CRE, despite having a very low cloud base (with high temperatures). Maybe some of Mark Zelinka's numerous publications on the topic may help.*
We removed this comment and presented the atmospheric longwave cloud radiative heating rates in the new Figure 10.

L315: *Also, you talk about the additional upper tropospheric warming due to climate change but never explain why should we care if the upper troposphere is slightly warmer? (compared to the arguably more important or at least more studied influence of changes in high clouds on the TOA radiative budget and climate sensitivity).*
We noted that an increase in atmospheric longwave heating from larger ICNCs will stabilise the atmospheric column and suppress deep convection.

L315-316: *I think the sentence "thicker cirrus..." is incorrect. Why only thick cirrus? Also, most cirrus aren't optically very thick.*
This sentence has now been removed.

L319-320: *I am not sure if the interpretation of the result of Sanderson et al., 2008 is correct, so it may need to be rewritten. Sanderson et al., 2008 found the IC fall speed to be important in modulating the mainly LW cloud feedback (and hence climate sensitivity) not because the IC fall speed would change between the present day and global warming simulation (IC fall speed is not calculated interactively in their simulations, given the use of a tuning parameter). However, a smaller ice fall speed leads to more high clouds. That in turn leads to a larger LW altitude (positive) cloud feedback, which is the dominant high cloud feedback. On the other hand, a smaller present-day cloud fraction due to large ice fall speed, leads to less high clouds and a smaller high cloud feedback and smaller climate sensitivity.*
We thank the Referee for drawing our attention to the details of the Sanderson et al. study. As noted above, we modified the discussion of such a "decreased-fall-speed" mechanism and removed the reference to Sanderson et al 2008. We agree that the smaller ice fall speed would lead to larger high-cloud fraction and a LW cloud feedback. Since we see instead decreased high-cloud fraction, such a mechanism is not dominant in our simulations.

Sec. 4.4.2: *As you talk about ice clouds and not only cirrus, you may want to also explore/mention the cloud phase negative optical feedback due to global warming (Tan et al., 2016, maybe also Bodas-Salcedo 2018 and 2019, Lohmann and Neubauer, 2018).*
We appreciate this suggestion but consider it outside the scope of this work. We would need to dedicate much more space and analysis to looking at shifts in overall cloud water and cloud water tendencies as well.

Concl:   *It may be appropriate to think a bit more about some of the questions I listed below and include some of that in the discussion: What did you learn about the model by exposing the number tendencies that you couldn't by simply taking the ICNC fields? Is there something that we should be worried about? Why? What is causing it? What are the potential weaknesses of the study? How does this compare to other work (if any exists – maybe for mass rates)?*

We enlarged and strengthened the Conclusions with additional discussion about the new analysis and the questions raised by the Referee.

**References**

[1] Bacer, S., Sullivan, S. C., Karydis, V. A., Barahona, D., Krämer, M., Nenes, A., Tost, H., Tsimpidi, A. P., Lelieveld, J., and Pozzer, A.: Implementation of a comprehensive ice crystal formation parameterization for cirrus and mixed-phase clouds in the EMAC model (based on MESSy 2.53), Geoscientific Model Development, 11, 4021–4041, 2018.

[2] Heymsfield, A. J., Krämer, M., Luebke, A., Brown, P., Cziczo, D. J., Franklin, C., Lawson, P., Lohmann, U., McFarquhar, G., Ulanowski, Z., and Van Tricht, K.: Cirrus Clouds, Meteorological Monographs, 58, 2.1–2.26, 2017.

[3] Krämer, M., Rolf, C., Luebke, A., Afchine, A., Spelten, N., Costa, A., Meyer, J., Zöger, M., Smith, J., Herman, R. L., Buchholz, B., Ebert, V., Baumgardner, D., Borrmann, S., Klingebiel, M., and Avallone, L.: A microphysics guide to cirrus clouds Part 1: Cirrus types, Atmospheric Chemistry and Physics, 16, 3463–3483, 2016.

[4] Muench, S. and Lohmann, U.: Developing a cloud scheme with prognostic cloud fraction and two moment microphysics for ECHAM-HAM, Journal of Advances in Modeling Earth Systems, 12, 2020.

[5] Sourdeval, O., Gryspeerdt, E., Krämer, M., Goren, T., Delanoë, J., Afchine, A., Hemmer, F., and Quaas, J.: Ice crystal number concentration estimates from lidar–radar satellite remote sensing – Part 1: Method and evaluation, Atmos. Chem. Phys., 18, 14327–14350, 2018

[6] Phillips, V. T. J., DeMott, P. J., and Andronache, C.: An Empirical Parameterization of Heterogeneous Ice Nucleation for Multiple Chemical Species of Aerosol, Journal of the Atmospheric Sciences, 65, 2757–2783, 2008.

---

## Author Comment (AC2) · 27 Nov 2020

**Authors' reply to Referee #2**

We thank the anonymous Referee #2 for the helpful comments. Below, we provide our replies to each comment; number of lines and sections refer to the old manuscript.

**Major Comments**

*(1) The model result uncertainty could be very large from a few aspects.*
*(1.1) The model grid spacing is very coarse (300 km) and the output time frequency is very sparse (every 5 hours). Many times, the cloud lifetime can be even less than 10 hours, then the sampling cannot be representative with every 5-hour time frequency. I'd suggest look at the sensitivity to model resolution (such as 100 km) and output time frequency (hourly) to meet the goal of quantification.*

We agree with the observations raised by the Referee. It would be interesting to perform sensitivity runs and investigate the influence of spatial and temporal resolutions on the tendencies. However, running new simulations at various resolutions with hourly output frequency would require much time, and new analysis should be performed. This is not the objective of this paper and could be addressed as an independent study. We mentioned at the end of the Conclusions that this can be an interesting future study.

*(1.2) Need to do ensemble runs for quantification.*

Also in this case, running ensemble experiments would require much time. Nevertheless, in the revised manuscript, the simulations were run for five years (instead of one year) so the analysis of the tendencies is now more robust.

*(1.3) Need to discuss that the results might be changed with different models or different physical parameterizations such as cumulus or microphysics parameterizations.*

In this regard, we added some new lines at the end of the Subsection 4.4.1, where we already discussed the sensitivity of the results to microphysics parameterization changes, and also in the Conclusions.

*(2) For the sink of ice crystal, sublimation should be considered.*

Unfortunately, as replied to Referee #1 point *II)*, the sublimation term is combined with SEDI; the separation between sublimation and sedimentation is not straightforward, and we cannot estimate the sublimation impact individually.

*(3) Result section: I feel a little surprised that the authors started the discussion of results for the source and sink of ice directly. It would be nice to understand the overall model performances in simulating radiation, clouds and precipitation first. Then get to the analysis of ice crystal number concentrations and its budget.*

We added a new section (4 "*Model results and evaluation of ICNC*") in the revised manuscript to evaluate the model ICNC against satellite ICNC retrievals before starting with the analysis of the tendencies.
The understanding of the overall model performance in simulating radiation, clouds and precipitation goes beyond the scope of this paper. The EMAC model is continuously developed, tested, and evaluated (against observations and other model results). The EMAC model and all its improvements are well documented in papers of the Special Issue "The Modular Earth Submodel System" of Copernicus and in the MESSy Consortium Website (https://www.messy-interface.org). Section 2.1 provides the standard description of EMAC; L94-95 cites some of the studies which deal with the model performance in simulating different physical quantities (e.g. aerosol burdens, cloud cover, radiation, cloud radiative effects...).

*(4) Since one of the purposes of the study is to test the sensitivity to two other nucleation parameterizations, then some description about the two default and two tested schemes is needed, particularly about how different they are in terms of representing ice formation such as temperature dependent, supersaturation dependent, and aerosol dependent.*
*If aerosol dependent, then what aerosols are considered? Why did you replace the immersion freezing scheme with a contact freezing scheme? Shouldn't they be considered together?*

The differences between the ice nucleation schemes in cirrus regime and mixed-phase regime are detailed in Bacer et al. 2018 (in Sections 2.2., 2.3.1, and Figure 1). We added some information regarding the schemes and also the reference. We specified at L138 that the parameterizations for heterogeneous nucleation are aerosol dependent. The ice nucleation parameterizations working in the mixed-phase regime are listed at L135-138: immersion freezing is not replaced with contact freezing; contact nucleation is always considered via LD06; immersion nucleation can be simulated either via LD06 or P13 (which also simulates deposition nucleation). We made L135-138 clearer.

**Minor Comments**

1. *Calling everything below -35 deg C as "cirrus clouds" is not accurate. I would suggest change to "pure ice clouds".*
   According to the definitions provided, for example, by Krämer et al. 2016 and Heymsfield et al. 2017, and the terminology used in most of the literature, we consider "cirrus clouds" (i.e. clouds purely composed of ice crystals) equivalent to "pure ice clouds", and we would like to keep this terminology in the manuscript.

2. *For the convective detrainment, does the model treat the detrainment at the levels with T> -35 deg C? If not, is there a reason? Theoretically convective detrainment of droplet and ice can occur from middle to top troposphere.*
   Convective detrainment can occur also at $T > -35°C$: the cloud condensate at $T < -35°C$ is considered in the ice phase (and it is a source of ICs), while the cloud condensate at $T > -35°C$ is considered in the liquid phase (and it is a source of cloud droplets). This is explained at L121-122.

3. *Line 210-215, does FREE include the droplet freezing in convective parameterization?*
   FREE does not include ice crystals formed in convective pararmeterizations. FREE is an independent term defined in the CLOUD submodel (convection is simulated by another submodel, CONVECT), and it includes the ICs formed from liquid water droplets that are transported in regions where temperature is $< -35°C$, as written at L163.

4. *Section 4.2, how to reconcile that DETR is much larger than NCIR in zonal mean (Fig. 2) but smaller than it in global spatial distribution (Fig. 1)?*
We are not sure what the Referee means here, as both Figure 1 and Figure 2 show that DETR is generally higher than NCIR, so the Figures are in agreement.

5. *Line 284-286, I am confused by this sentence. Earlier it is said LD06 is a contact freezing scheme which is for heterogenous freezing. Here you said LD06 parameterizes only homogeneous nucleation. Also P13 should be an immersion freezing scheme which should be much more efficient than the contact freezing LD06, but the results in section 4.4.1 did not even mention the differences they can make.*
We thank the Referee for noticing that there is indeed an inconsistency at L285; we replaced *"LD06"* with *"KL02"*.
Since NCIR and NMIX are defined as the rates of new ICs in the cirrus regime and new ICs in the mixed-phase regime, it is not possible to discern the contributions from contact and immersion freezing. However, during some previous tests, we found that immersion nucleation simulated with LD06 produces more ICs than immersion-condensation and deposition nucleation using P13. This is in agreement with Phillips et al. 2008, who compared their empirical parameterization (which is the previous version of P13) with other parameterizations including LD06.

**References**

[1] Bacer, S., Sullivan, S. C., Karydis, V. A., Barahona, D., Krämer, M., Nenes, A., Tost, H., Tsimpidi, A. P., Lelieveld, J., and Pozzer, A.: Implementation of a comprehensive ice crystal formation parameterization for cirrus and mixed-phase clouds in the EMAC model (based on MESSy 2.53), Geoscientific Model Development, 11, 4021–4041, 2018.

[2] Heymsfield, A. J., Krämer, M., Luebke, A., Brown, P., Cziczo, D. J., Franklin, C., Lawson, P., Lohmann, U., McFarquhar, G., Ulanowski, Z., and Van Tricht, K.: Cirrus Clouds, Meteorological Monographs, 58, 2.1–2.26, 2017.

[3] Krämer, M., Rolf, C., Luebke, A., Afchine, A., Spelten, N., Costa, A., Meyer, J., Zöger, M., Smith, J., Herman, R. L., Buchholz, B., Ebert, V., Baumgardner, D., Borrmann, S., Klingebiel, M., and Avallone, L.: A microphysics guide to cirrus clouds Part 1: Cirrus types, Atmospheric Chemistry and Physics, 16, 3463–3483, 2016.

[4] Muench, S. and Lohmann, U.: Developing a cloud scheme with prognostic cloud fraction and two moment microphysics for ECHAM-HAM, Journal of Advances in Modeling Earth Systems, 12, 2020.

[5] Sourdeval, O., Gryspeerdt, E., Krämer, M., Goren, T., Delanoë, J., Afchine, A., Hemmer, F., and Quaas, J.: Ice crystal number concentration estimates from lidar–radar satellite remote sensing – Part 1: Method and evaluation, Atmos. Chem. Phys., 18, 14327–14350, 2018

[6] Phillips, V. T. J., DeMott, P. J., and Andronache, C.: An Empirical Parameterization of Heterogeneous Ice Nucleation for Multiple Chemical Species of Aerosol, Journal of the Atmospheric Sciences, 65, 2757–2783, 2008.

---

## Author Response (AR2)

**Authors' reply**

December 22, 2020

Dear Editor,

we thank you for your comments. Below, we reply to your comments point-by-point. At the end of the document, we list other changes made in the last version of the manuscript.

**Minor revisions**

1. *How do your findings relate to models with higher spatial resolution? Clearly, running simulations at higher resolution is beyond the scope of this revision but would you expect that observed numerical tendencies also occur at a higher resolution scale?*
   We think that most of our results would be qualitatively similar at higher resolution. The numerical tendencies could become even more relevant at higher resolution, as local gradients may become steeper such that discrepancies requiring numerical tendencies will be maintained or even increased. A stronger impact may result from a change in vertical resolution as the microphysics is more susceptible to changes in vertical motion and associated thermodynamics.

2. *It is not clear why sublimation as a sink cannot be treated independently from sedimentation. Maybe add a sentence on this matter to make this clearer.*
   The point is that sublimation of ICs is not explicitly treated in CLOUD. If cloud ice sublimates, the ice mass is reduced to zero; consequently, the ice number concentration is also reduced to zero by a numerical tendency. In case of incomplete sublimation, all ICs shrink, and ICNC does not change. In the end, it is not correct saying that sublimation is considered in the sedimentation term as it could be part of a numerical tendency. We removed this part at L167 and wrote "as it is not explicitly treated."

3. *Conclusions: It should be pointed out that sublimation is omitted in the analysis, given that sublimation is a substantial sink of upper tropospheric ice.*
   We modified L420: "The loss processes of ice crystals are melting (MELT), self-collection (SELF), aggregation (AGGR), and accretion (ACCR); sublimation is excluded from the analysis of this study."

4. *Abstract: Are the ICNC in good agreement with satellite observations? Is this statement accurate? Figures 1 and 2 do not corroborate this clearly. Maybe use same colormap for Figs. 1 and 2 to make it easier for the reader to interpret both figures.*
   We modified L8-9 in the Abstract to make it more precise: "We found that model ICNCs are in overall agreement with satellite observations in terms of spatial distribution, although the values are overestimated, especially around high mountains."
   We agree with the Editor that using the same color bar would facilitate, in principle, the comparison. However, we found that this was not the case for the comparison between DARDAR and EMAC as the DARDAR plots lost clarity in their patterns using a discrete color bar and increasing the maximum value (to match the one of the EMAC color bar). Therefore, we believe that Figs 1 and 2 are more representative with the current color bars.

5. *End of Section 4: It would be valuable to mention here that the NOfree simulation compares well with observations. That would also give an additional argument for examining NOfree in section 5.*
   We added a new sentence at L252: "Interestingly, the ICNC zonal mean computed with the NOfree

simulation (not shown) is closer to the observations with respect to the REF simulation, suggesting that the FREE tendency contributes to the overestimation of ICNC."

6. *Page 13, line 296: DETR is independent of updrafts? Why, then, is DETR higher over land?*
DETR depends on the detrained mass flux, which depends on the convective mass flux or pressure velocity and ultimately vertical velocity. These vertical velocities are higher over land because the thermal structure is favorable for generation of larger buoyancies [Del Genio et al. 2007]. In EMAC, DETR is computed on the basis of the cloud condensate parameterized by the CONVECT submodel, which simulates convective clouds and does depend on updrafts as written at L123: "Convective cloud microphysics in EMAC is solely based on temperature and updraft strength...".

7. *Page 16, line 310-312: It does not look like NCIR is the dominant source of ICs between 350 and 200 hPa. Based on Fig. 6, it seems to dominate only above approximately 200 hPa level.*
We agree that the sentence was not correct; we made the sentence more specific: "We clearly see that ice nucleation in the cirrus regime (NCIR) is the dominant source of ICs at pressures lower than 200 hPa between the tropics and lower than 350 hPa at high latitudes."

8. *Page 20, line 397: Did you mean homogeneous freezing level instead of melting level?*
Yes, thank you for noticing this; we corrected the sentence.

9. *Page 20, lines 406-407: Does the ice water content change between FUT and REF simulations?*
The global means of ice water path (IWP) in REF, $(12.71 \pm 0.27)\mathrm{g/m}^2$, and in FUT $(12.13 \pm 0.25)\mathrm{g/m}^2$, are very close; the zonal mean of ice water content (IWC) in FUT (not shown) increases in the upper troposphere from the homogeneous freezing level. Since the entire paper focuses on ICNC, we did not include this information in the manuscript and we deleted the sentence at L410-412.

10. *Page 21, line 434-435: "consists in" reads a bit awkward. Also, please elaborate on the statement "FREE should not depend only on CDNC". On what does it depend?*
Thank you for noticing this; we replaced it with "consists of".
At L433, we meant that FREE could depend also on updraft velocity like in Muench and Lohmann (2020), as we wrote at L430. We made this paragraph clearer in the manuscript.

11. *Page 21, line 457: Did you refer to the homogeneous "freezing level"?*
Yes. We changed "freezing level" to "homogeneous freezing level" also in other parts of the manuscript to be clearer.

12. *Page 21, line 458: You can omit "vertical".*
We removed "vertical".

**Further changes**

- [L330-335] "It must be stressed that the IC sources and sinks of Figures 6 and 7 cannot be expected to balance for *the following* reasons. First, the tendencies of physical processes are not computed in this study, i.e. transport due to advection, turbulence, and convection and sedimentation ($R_{transp}$ and $R_{sedi}$ in equation (1), respectively). In particular, $R_{transp}$ is not computed in the CLOUD submodel but derives from various submodels in EMAC, e.g. CVTRANS (Tost et al., 2010) and E5VDIFF (Roeckner et al., 2004). Second, *sublimation is a missing sink in this study. Finally,* numerical tendencies also affect ICNC at each model time step and play a significant role in the ICNC budget (as discussed in Subsection 5.1)."

- [Section 4] We specified that the model results in Figs. 1 and 2 refer to the REF simulation.

- [L468] "The *tendencies of transport, sedimentation, and sublimation* could also be included to close the ICNC budget."

---

## Author Response (AR3)

**Authors' reply**

December 27, 2020

Dear Editor,

thank you very much for your comment. We replaced "homogeneous freezing level" with "homogeneous nucleation freezing level" (at L395, L411, and 457).

Yours faithfully,

Sara Bacer